# Growth-rate-dependent and nutrient-specific gene expression resource allocation in fission yeast

Istvan T Kleijn[1,2,3,*], Amalia Martínez-Segura[1,2,*], François Bertaux[1,2,3], Malika Saint[1,2], Holger Kramer[1,2], Vahid Shahrezaei[3], Samuel Marguerat[1,2]

Cellular resources are limited and their relative allocation to gene expression programmes determines physiological states and global properties such as the growth rate. Here, we determined the importance of the growth rate in explaining relative changes in protein and mRNA levels in the simple eukaryote *Schizosaccharomyces pombe* grown on non-limiting nitrogen sources. Although expression of half of fission yeast genes was significantly correlated with the growth rate, this came alongside wide-spread nutrient-specific regulation. Proteome and transcriptome often showed coordinated regulation but with notable exceptions, such as metabolic enzymes. Genes positively correlated with growth rate participated in every level of protein production apart from RNA polymerase II–dependent transcription. Negatively correlated genes belonged mainly to the environmental stress response programme. Critically, metabolic enzymes, which represent ~55–70% of the proteome by mass, showed mostly condition-specific regulation. In summary, we provide a rich account of resource allocation to gene expression in a simple eukaryote, advancing our basic understanding of the interplay between growth-rate-dependent and nutrient-specific gene expression.

## Introduction

Cellular growth is the process by which cells increase in mass. It is a fundamental systemic process that impacts most aspects of cell physiology. Growth can be very fast, for example, yeast cells can double in mass every few hours, and certain bacteria only require minutes. Conversely, slower growth is observed in multicellular organisms, in which several cell types take days to grow and divide. Crucially, the cellular growth rate changes in response to external cues such as nutrient quality, stressing agents, or growth factors.

Measurements of biomass composition in unicellular organisms have long-established cellular growth rates as a covariate of cell physiology (Schaechter et al, 1958; Mitchison & Lark, 1962; Waldron & Lacroute, 1975; Fantes & Nurse, 1977; Neidhardt et al, 1990; Bremer & Dennis, 2008). In the last decade, quantitative experimental work, together with mathematical modelling, have described this relationship (reviewed in Klumpp and Hwa [2014], Shahrezaei and Marguerat [2015], Jun et al [2018], and Bruggeman et al [2020]). This body of work has emphasised how the macromolecular composition of the cell is tightly connected to growth rate. Specifically, for cultures undergoing balanced exponential growth modulated by external nutrients, the total RNA abundance per unit of biomass and the growth rate are correlated linearly. This phenomenological relationship is called the first or ribosomal growth law and reflects an increased requirement for ribosomes during faster growth to support protein synthesis. The demand for ribosomes is also felt at the protein level, where it induces a trade-off between proteins involved in translation and those involved in catabolism. It was shown that about half of the total protein mass in *Escherichia coli* responded to growth modulations by nutrient limitation and translational inhibition (Scott et al, 2010; You et al, 2013). These observations were formalised in a phenomenological model separating the proteome into three broad sectors based on their growth rate correlations. Proteins showing expression levels positively correlated with the cellular growth rate during nutrient limitation and negatively during translational inhibition form the R-sector (as many of them are constituents of the ribosome). Conversely, proteins with expression levels negatively correlated with the growth rate form the P-sector. Proteins that do not respond to the growth rate belong to the Q-sector (Scott et al, 2010). The concept of proteome sectors has been the basis of several phenomenological and coarse-grained mechanistic models relating optimal resource allocation to protein abundance and cellular growth rates (Molenaar et al, 2009; Scott et al, 2014; Maitra & Dill, 2015; Weiße et al, 2015; Pandey & Jain, 2016; Liao et al, 2017; Bertaux et al, 2020; Hu et al, 2020).

The molecular mechanisms behind the phenomenological assignment to the three proteome sectors remain less well understood.

[1]Medical Research Council London Institute of Medical Sciences (MRC LMS), London, UK   [2]Institute of Clinical Sciences (ICS), Faculty of Medicine, Imperial College London, London, UK   [3]Department of Mathematics, Faculty of Natural Sciences, Imperial College London, London, UK

Correspondence: v.shahrezaei@imperial.ac.uk; samuel.marguerat@ucl.ac.uk
François Bertaux's present address is Institut Pasteur, Paris, France.
Samuel Marguerat's present address is UCL Cancer Institute, University College London, London, UK.
*Istvan T Kleijn and Amalia Martínez-Segura contributed equally to this work.

R-sector proteins are universally involved in translation and ribosome biogenesis and many of them are controlled by global signalling pathways such as guanosine tetraphosphate (ppGpp) in prokaryotes or the target of rapamycin complex 1 (TORC1) in eukaryotes (Irving et al, 2020; Petibon et al, 2020). P-sector proteins, on the other hand, are more diverse and often involved in metabolic adaptation and stress response (Brauer et al, 2008; You et al, 2013; Hui et al, 2015; Schmidt et al, 2016). In *E. coli*, the master regulator CRP-cAMP has been proposed to control the P-sector assignments of carbon catabolism enzymes when growth rate was modulated by the quality of abundant carbon sources (You et al, 2013). Under other growth modulations and in other organisms, whether the regulation of P-sector proteins is as directly mechanistically linked to the growth rate as for R-proteins is less clear.

Transcriptomics and proteomics have been instrumental in characterising the coordination between gene expression and cellular growth. The ribosomal growth law was first confirmed in the *E. coli* proteome in continuous cultures limited by carbon availability (Peebo et al, 2015), under titrations of carbon, nitrogen, and translational inhibition (Hui et al, 2015), and in an extensive study of 22 growth conditions (Schmidt et al, 2016). In addition, the Hui study proposed that the P-sector could be divided into subsectors related to different metabolic functions depending on the type of nutrient limitation. In the budding yeast *Saccharomyces cerevisiae*, a seminal microarray study showed strong correlations between hundreds of transcripts with the chemostat dilution rate across six nutrient titrations (Brauer et al, 2008). The observed correlations agreed with the ribosomal growth law and highlighted stress response as a component of the P-sector alongside metabolic functions. More recently, Metzl-Raz et al (2017) observed the ribosomal growth law in the proteome of budding yeast after combining existing data sets of cultures grown in a variety of carbon sources (Paulo et al, 2015, 2016) with data obtained under nitrogen and phosphorus limitation (Metzl-Raz et al, 2017). They also proposed that a pool of non-translating ribosomes is available as a buffer during changing growth conditions, a strategy also observed in prokaryotes (Dai et al, 2016; Mori et al, 2017; Kohanim et al, 2018). This suggests that resource allocation may not be fully optimised for maximal cell growth. Signs of excess capacity have also been reported for metabolic pathways, including glucose catabolism (Yu et al, 2020). Further omics studies in *S. cerevisiae* have defined additional characteristics of resource allocation such as reallocation of proteome mass from amino acid biosynthesis to protein translation upon amino acid supplementation (Björkeroth et al, 2020), or the respective contribution of transcription and translation to different allocation strategies (Yu et al, 2021). Thus, genome-wide omics experiments have been key to improve our understanding of resource allocation in *E. coli* and *S. cerevisiae* by connecting proteome sectors to specific physiological functions.

The cellular growth rate reflects the metabolic state of the cell and in limiting nutrient conditions metabolic enzymes are often part of the P-sector (Hui et al, 2015; Schmidt et al, 2016). This suggests that expression levels of specific metabolic enzymes when responding to external conditions can be directly regulated alongside the growth rate. The cell metabolism however is an exquisitely complex network of interconnected processes and perturbation of single pathways can have wide-spread systemic effects. Central carbon metabolism (CCM) relies on three pathways: glycolysis, the pentose phosphate pathway, and the tricarboxylic acid (TCA) cycle. Together, these generate energy in the form of ATP, in a process mediated by reducing agents such as NADH, and produce building blocks for biosynthesis. ATP can be generated anaerobically via fermentation; a process which consists of glycolysis and the subsequent degradation of pyruvate, or aerobically via respiration, which requires the TCA cycle and subsequent oxidative phosphorylation (OXPHOS). The extent of fermentative versus respiratory metabolism affects the NAD+/NADH redox balance and vice versa, as NAD+ reduction during glycolysis and the TCA cycle must be balanced by NADH oxidation occurring during pyruvate degradation and OXPHOS (Vemuri et al, 2007; van Hoek & Merks, 2012; Campbell et al, 2018; Luengo et al, 2020). In eukaryotes, these reactions are compartmentalised between the cytoplasm and the mitochondria, with the latter housing the respiratory enzymes and functioning as hubs that connect diverse metabolic pathways including CCM and amino acid metabolism (Spinelli & Haigis, 2018). For instance, amino acid degradation enables the assimilation of nitrogen as ammonium or glutamate via de- or transamination reactions. The remaining carbon backbone is recycled into the cell's biomass or excreted, and the associated metabolites affect carbon metabolism (Godard et al, 2007). Importantly, mitochondrial intermediates are required for amino acid biosynthesis even during fermentative energy generation (Malecki et al, 2020). In fission yeast, a single point mutation in the pyruvate kinase Pyk1, affecting its activity, has been shown to rebalance the fluxes through the fermentation and respiration pathways alongside shifts in the transcriptome and proteome composition (Kamrad et al, 2020), giving a prime example of how the cell co-adjusts perturbations in metabolic fluxes and expression burdens. Taken together, shifts in the metabolic demand propagate throughout the cell, as most metabolic pathways are tightly interlinked (Chubukov et al, 2014).

The expression levels of CCM enzymes, and therefore the fluxes through the pathways depend on external conditions and stress levels. As a result, cellular states and metabolic strategies are linked to resource allocation to different gene expression programmes. For example, during rapid growth on glucose, yeast utilises the fermentative pathway alongside the TCA cycle even in the presence of oxygen, a phenomenon known as aerobic glycolysis or the Crabtree effect (Shimizu & Matsuoka, 2018). Aerobic glycolysis is also a characteristic of tumour cells, for which it is known as the Warburg effect (Vander Heiden et al, 2009). This strategy appears counterintuitive as fermentation generates fewer molecules of ATP per glucose molecule than respiration. Several hypotheses have been proposed to resolve this paradox. All require a second growth-limiting constraint besides glucose uptake which would be specific to respiro-fermentative growth (de Groot et al, 2019). Examples include the cytoplasmic density of macromolecules (Vazquez et al, 2008; Goelzer et al, 2015), total proteome allocation (Basan et al, 2015), and membrane area availability (Szenk et al, 2017). Thus, a whole-cell understanding of cellular trade-offs between multiple constraints must take into account gene expression alongside metabolic maps (Goelzer & Fromion, 2017; Yang et al, 2018; Dahal et al, 2020). Resource allocation constraints have been successfully introduced into genome-wide metabolic models of several organisms as more high-quality expression data hane become available (O'Brien et al, 2013; Sánchez et al, 2017; Chen et al, 2021). In summary, quantitative surveys of the

gene expression cost of metabolic pathways are key to understanding cell physiology.

Here, we define the growth-rate-dependent and nutrient-specific resource allocation to the fission yeast *Schizosaccharomyces pombe* proteome and transcriptome. We find that both types of regulation are interconnected and define protein synthesis and stress response as the processes positively and negatively regulated with the growth rate. We then study the plasticity of the gene expression burden of metabolic pathways in response to changes in nutrients and their reliance on transcriptional and post-transcriptional regulation. Altogether we provide a rich account of resource allocation in a simple eukaryote as a function of external conditions.

# Results

### Fission yeast gene expression shows growth-rate-related and condition-specific components

To generate cell populations that grow at different rates but not limited for nutrients, we used eight defined culture media, each containing a unique source of nitrogen. *S. pombe* 972 h$^-$ prototroph wild-type cells were grown in turbidostats at constant concentrations of $OD_{600}$ ~ 0.4 (3–5 × 10$^6$ cells/ml) in triplicates (Fig 1A). Like in chemostats, turbidostat cultures are diluted by the addition of fresh medium. In the case of the turbidostat system, however, it is the cell concentration that is directly measured and maintained constant and not the proliferation rate. This ensures that cellular growth is not limited by a lack of nutrients, but rather determined by the quality of the provided nitrogen source and the resulting internal allocation patterns. Growth rates were measured halfway through the procedure during a twofold dilution cycle (Fig 1B and Table S1). Cells were then let to reach the target $OD_{600}$ again, grown for a second period of stable $OD_{600}$ to ensure that they were at steady state, and harvested (Fig 1B and Table S1). The growth media have been extensively characterised elsewhere (Fantes & Nurse, 1977; Carlson et al, 1999; Petersen & Russell, 2016). Seven media contained 20 mM of a single amino acid (Trp, Gly, Phe, Ser, Ile, Pro, and Glu), and one 93.5 mM of ammonium chloride ($NH_4Cl$, referred to as Amm) as a reference. In our hands, this design achieved growth rates ranging 0.05–0.28 h$^{-1}$ for 43–143 h (6–28 generations depending on the nitrogen source) (Fig 1C–E and Table S1) (Takahashi et al, 2015). To measure the proteome and transcriptome allocation as a function of the growth rate, we performed label-free proteomics and RNA sequencing (RNA-Seq) analysis of cells from each culture condition (see the Materials and Methods section and Tables S2–S5).

We first asked whether the fission yeast proteome composition differed significantly between the eight growth conditions. Strikingly, ~45% of the 1,988 protein groups robustly detected in all samples were significantly more variable across conditions than among biological replicates (Holm-adjusted $P_{ANOVA}$ < 0.05). This pervasive level of gene regulation was also apparent at the transcriptome level where ~52% of mRNAs showed significant variability. These results indicate that the composition of the proteome and transcriptome are both strongly affected by conditions that change the growth rate.

To investigate this variability further, we used the *z*-score transformed protein fraction of each gene for hierarchical clustering

(Fig 2A, see the Materials and Methods section). This treatment enabled normalisation for protein expression levels across the proteome while preserving the variation of each protein between conditions. We defined 10 clusters that revealed two major features of the data sets (Fig 2A–C). First, all clusters showed clear differences in protein expression across one or more conditions. Second, the expression of several proteins was not strictly condition-specific but instead showed a coordinated linear increase with growth rate (clusters 7 and 8). Interestingly, the total baseline expression of the condition-specific clusters was positively (clusters 7, 8, and 9), or negatively (clusters 1, 2, 3, and 6) correlated with the growth rate. Apart from clusters 1, 4, and 10, clusters were enriched for defined functional categories, indicating that the shifting balance between condition-specific regulation and growth rate regulation may have physiological consequences related to the enriched functions (Figs 2A and S1).

Both modes of regulation were also apparent in the transcriptome data for coding (Figs 2C and S2A and B) and non-coding RNA (ncRNA) (Fig S3A and B). Interestingly, most ncRNAs showed clear and reproducible condition-specific expression between replicates, suggesting the presence of active regulation, consistent with analyses using different genetic and physiological conditions (Atkinson et al, 2018). To test this hypothesis, we compared the expression patterns of ncRNA from each cluster with the expression of their flanking coding genes (Fig S3C and D). We found that, apart from the growth-rate-correlated cluster 1, expression of individual ncRNAs was not mirrored by expression of their neighbouring mRNAs. This indicates that many ncRNA are subjected to some level of independent regulation. In summary, we find that regulation of gene expression programmes across conditions that affect the growth rate has two components; one which is condition-specific and another which is coordinated with growth rate.

### Growth-dependent gene expression is an important determinant of the cell protein and mRNA composition

We first focused our analysis on the growth-dependent component of fission yeast gene expression. Linear correlations between the expression of individual genes and the growth rate have been observed in several organisms under different types of growth limitation (Brauer et al, 2008; Hui et al, 2015; Peebo et al, 2015; Schmidt et al, 2016; Metzl-Raz et al, 2017; Zavřel et al, 2019). Following the terminology used in prokaryotes, we divided proteins and mRNA into three sectors depending on whether they show a growth-dependent component that was positively (R), negatively (P), or not significantly (Q) correlated with the growth rate (Scott et al, 2010, 2014). We used repeated-median linear models to quantify the linear coordination of each protein and mRNA quantity with growth. This model fits a linear dependence in the presence of large numbers of outliers and is therefore robust to the condition-specific component of gene expression (see the Materials and Methods section, Fig S4A–F and Table S6).

The linear fits generated two useful parameters. First, the slope of the linear regression is a measure of the strength of the dependence of a protein's concentration on the growth rate. Second, its *y*-intercept represents the fraction of the protein numbers that is not directly dependent on growth. Both parameters are directly correlated with expression levels making it difficult to disentangle

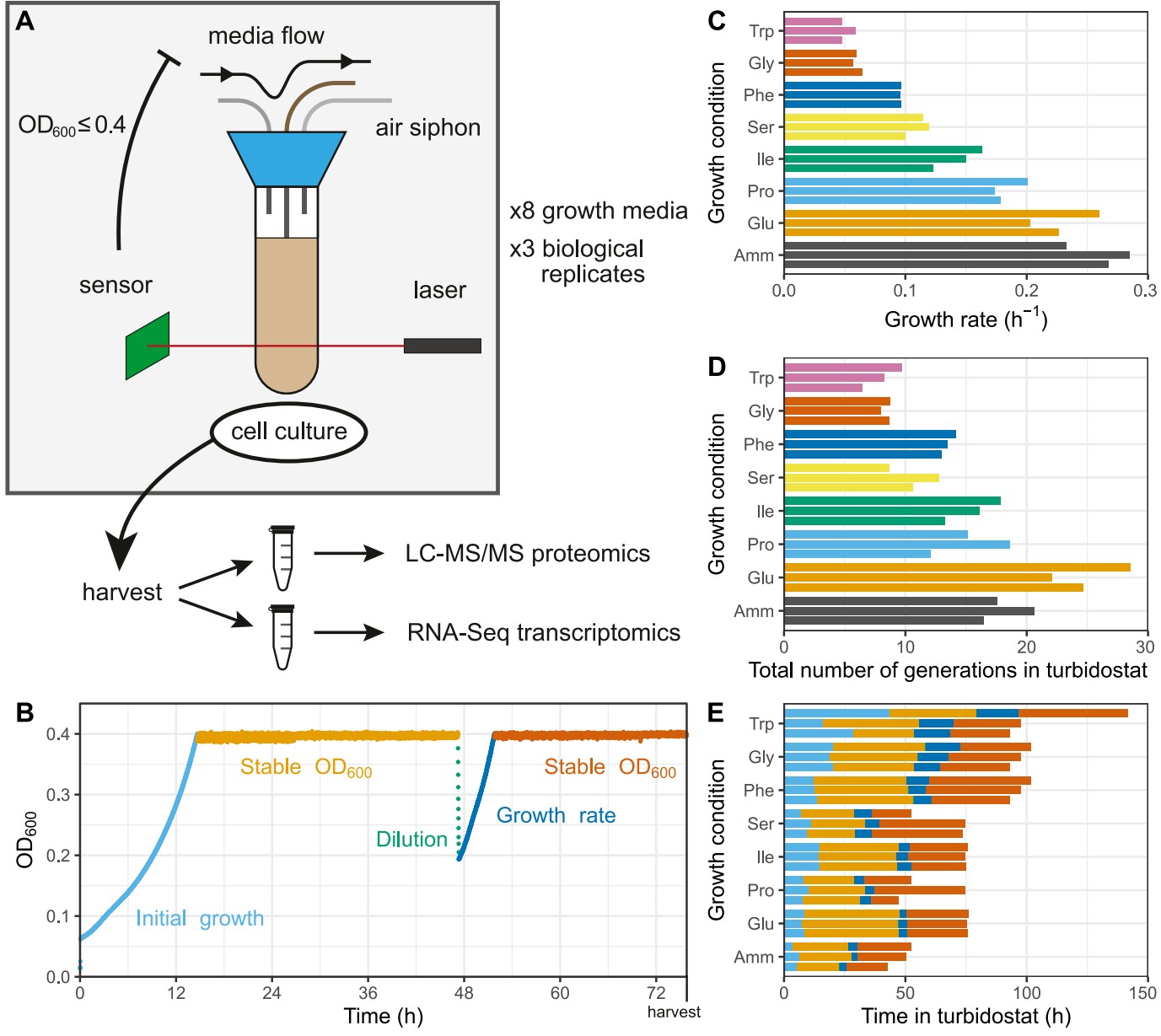

**Figure 1. Characterisation of culture growth in turbidostats across eight minimal media.**
**(A)** Illustration of the turbidostat culture chamber with the control flow and analysis pipeline. **(B)** Example growth curve (Ile replicate 2) showing different growth phases in the turbidostat. **(C)** Estimated growth rates $\mu$ based on a twofold dilution and regrowth cycle for the eight growth media using three biological replicates each. Amm, ammonium chloride, equal to standard EMM2 medium. **(D)** Total number of generations each culture spent in a turbidostat. **(E)** Total time in hours each culture spent in a turbidostat, with the duration of individual growth phases coloured as in B. Note that, with $N_G$ the number of generations, $T$ the time spent in the turbidostat, $\mu$ the growth rate, and $T_d$ the doubling time, $T_d = \ln(2)/\mu$ and $N_G = T/T_d$.

the strength of the growth-rate-related regulation from an mRNA or protein from its abundance. To take this into account, we developed a normalised measure of growth dependence, defined as the ratio of the difference in expression level between zero and maximum growth and the median expression. It therefore denotes the "fold change" or FC of growth rate changes relative to an intermediate baseline (see the Materials and Methods section, Fig S4G and H). FC values are a combination of the regression slope and $y$-intercept which do not scale with abundance, thereby enabling a direct

comparison of the growth dependence of single genes or groups thereof.

Repeated-median linear models captured the growth-dependent component of the 10 clusters from Fig 2, and proteins from the R- and P-sectors dominated the clusters that were positively and negatively correlated with growth, respectively (Fig S5). Of all the genes detected in the proteome across the eight conditions examined, we found that 22% of proteins and 37% of mRNA belonged to the R-sector; similarly, 24% and 21% of the proteins and mRNA

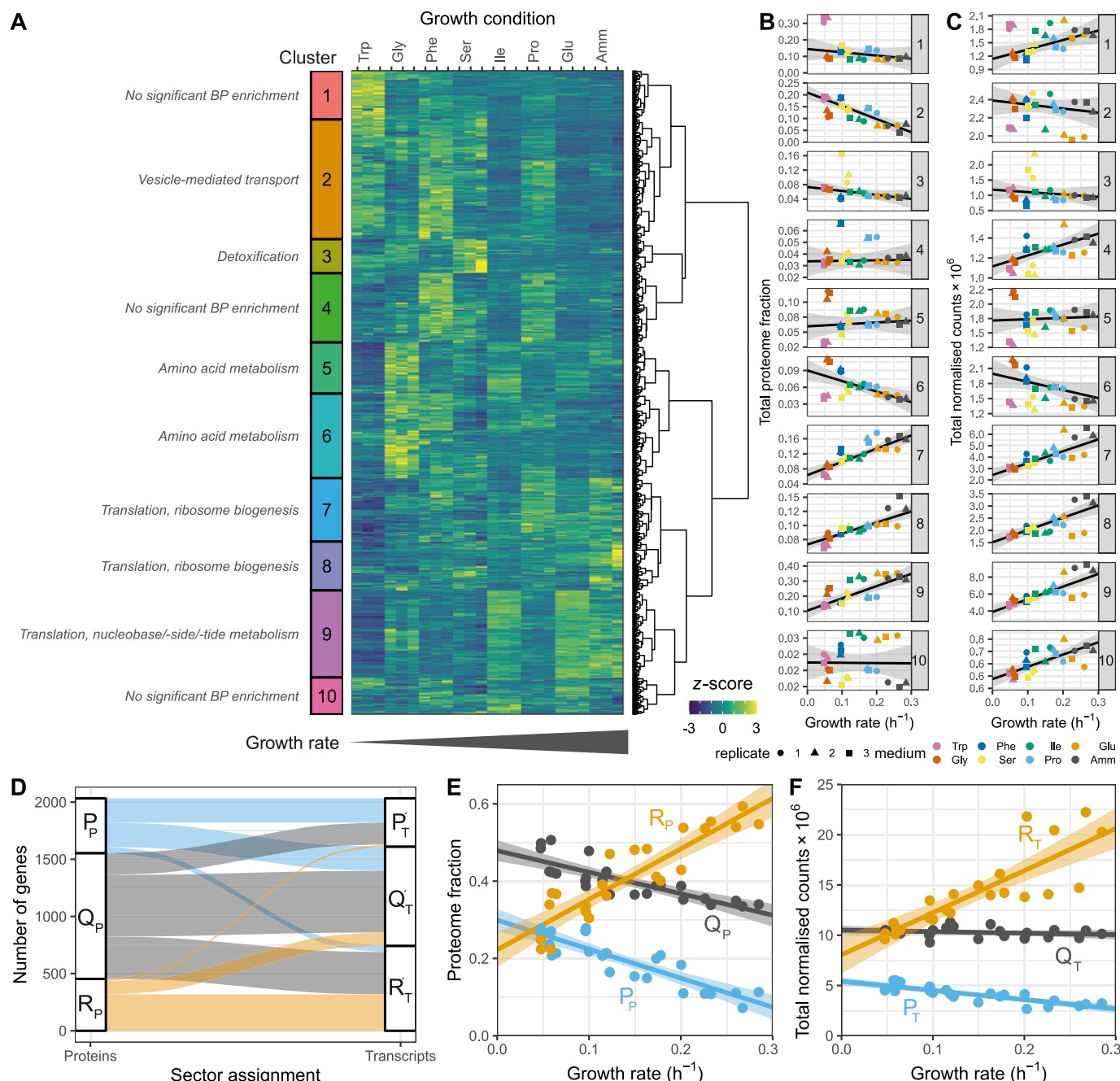

**Figure 2. Fission yeast gene expression shows growth-rate-dependent and nutrient-specific components.**
**(A)** Hierarchical clustering of *z*-score transformed protein expression fractions for the 1,988 protein groups detected across all conditions for cells grown in seven single amino acids or NH₄Cl (Amm) using three biological replicates. Growth conditions are ordered by increasing growth rate. 10 clusters are labelled on the left together with manual summary of enriched functional categories (see Fig S1). The colour scale was truncated at *z*-scores of −3 and +3. **(B)** Summed protein mass fractions for the 10 clusters defined in A as a function of the growth rate. Repeated-median linear model (RMLM) fit is shown as a black line and the predicted 2.5th–97.5th percentile confidence interval (CI) of the fit as the grey shaded area. **(C)** As shown in (B), for DESeq2-normalised RNA-Seq counts. **(D)** Assignment of 2,030 proteins detected across all conditions and their respective transcripts to the R (orange), P (blue), and Q (grey) sectors based on protein fractions (left) and DESeq2-normalised counts (right). Each protein is connected to its corresponding transcript by a line and colours are according to the protein sectors. **(E)** Sum of protein fractions for the R- (orange), P- (blue), and Q- (grey) sectors as a function of growth conditions. The figure includes all 3,498 protein groups detected in at least one condition. Best fit and predicted CI are plotted for the ordinary least squares linear model. **(F)** As shown in (E), for DESeq2-normalised RNA-Seq counts for 5,135 detected genes. Abbreviations: P_P, Q_P, R_P: protein groups assigned to P-, Q-, and R-sector. P_T′, Q_T′, R_T′: transcripts corresponding to protein groups detected across all conditions assigned to P-, Q-, and R-sector. P_T, Q_T, R_T: all transcripts assigned to P-, Q-, and R-sector.

belonged to the P-sector, respectively. The protein and mRNA of a given gene belonged to the same sector in 53% of the cases (Fig 2D). When they did not, the mRNA of P- or R-proteins were mostly assigned to the Q-sector and vice versa, with only 19 R-proteins having P-sector mRNA, and 55 P-proteins having R-sector mRNA, out of the 2,033 proteins detected.

In quantitative terms, the total proteome mass fraction of the fission yeast R-sector ranged between ~20% at zero growth and 55% for the fastest measured growth rate, whereas the mass fraction of the P-sector similarly ranged from ~30 to 10% (Fig 2E). The sum of all Q-sector proteins was negatively correlated with the growth rate because proteome fractions add up to one by definition. However, none of the individual proteins showed significant correlation with the growth rate. At the mRNA level, the R-fraction ranged from 38 to 59% of the total normalised counts, and the P-fraction from 19 to 10% (Fig 2F). Thus, during fast growth, over half of the gene expression burden is dedicated to factors that increase in concentration with growth rate and may therefore be limiting. Moreover, the amplitude of the variability in the concentration of fission yeast proteins and mRNA that depend on the growth rate alone is in the order of magnitude of the cut-offs that are commonly used for differential expression analysis. Therefore, differences in growth rate are important factors that affect interpretation of transcriptomics and proteomics data (Yu et al, 2021).

### Growth-dependent gene expression is preserved between mRNA and protein

Having performed both transcriptomics and proteomics data on the same cells enabled us to compare the two levels of gene expression in a unified data set. To perform a like-for-like comparison, we converted our expression measure in both data sets to relative number fractions (Balakrishnan et al, 2021 Preprint). First, we explored the correlation between protein and mRNA levels averaged across all genes, using the Spearman correction to account for the varying reproducibility of the data (Csárdi et al, 2015; Franks et al, 2017) (see the Materials and Methods section). Messenger RNA reliabilities were in the range of 97.5–99.8% and protein reliabilities were between 92.8 and 97.6% indicating high concordance between biological repeats (Figs S6 and S7 and Table S7). Spearman-corrected correlations of log-transformed relative protein and mRNA levels were ~0.8 for most conditions, with a slightly elevated correlation in EMM2 reference medium and slightly smaller correlation in Trp medium suggesting marginal medium-specific effects (Figs 3A and S8). Furthermore, we found evidence of post-transcriptional amplification: the ratio of protein to mRNA generally increased with protein expression, but a plateau was reached at very high expressions (Fig S9). This agrees with earlier observations (Marguerat et al, 2012). In summary, our analysis indicates that mRNA expression levels are globally good predictors of proteome composition in our system.

Second, we explored the extent of post-transcriptional regulation for each given gene in different growth media. For each gene, we calculated the $\log_2$-transformed ratio of protein and mRNA relative number fractions and subtracted from this the median ratio across conditions (Franks et al, 2017) (see the Materials and Methods section, Table S8). Subsequently, we performed hierarchical clustering analysis (Fig 3B) and a functional enrichment of

the clusters (Fig S10). There was little growth-dependent variation in the resulting residual protein-to-RNA ratios of many genes, including ribosomal proteins (RPs) (Fig 3B cluster 10, Fig S10). However, some genes showed signs of medium-specific post-transcriptional regulation, prominent in Trp, Ile, and Glu (clusters 1–6). Notably, clusters 4 and 5 contained genes with elevated protein-to-mRNA ratios in Ile and Glu, but repressed ratios in Pro, Ser, Phe, and (chiefly) Trp. The enrichment analysis highlighted a moderate enrichment for metabolism in cluster 5.

Next, we compared the size of the growth-rate-dependent effects between protein and mRNA by contrasting the fold change measures of genes present in both data sets. As shown in Fig 3C and in accordance with Figs 2D and S5, the RMLMs showed good agreement between the two types of data. Protein FC measures were generally larger than transcript FCs, again highlighting the post-transcriptional amplification. Further study of the disagreeing genes showed a minor enrichment of proteasomal genes in the group with negative growth rate correlations in the proteome and positive growth rate correlations in the transcriptome (Fig S11).

Finally, we compared mRNA and protein growth-related regulation for a series of functional categories using an unbiased gene set enrichment analysis, ranking genes on the signed significance measure used to determine the P- and R-sector (see the Materials and Methods section, Tables S9 and S10). This showed that most categories showed a similar growth-related regulation for both mRNA and proteins (Fig 3D). This finding is robust to changing the ranking variable to the effect size (FC) instead of the significance (Tables S11 and S12 and Fig S12B). Transcripts for transcription factors, and for proteins generally bound to the chromosome, were an exception. This was due to limited coverage in the proteomics data for these categories (Fig S12A). Specific functional categories will be discussed below.

### R-sector proteins participate in all steps of the protein synthesis process

We next queried the cellular processes that had a strong R component and could therefore be either limiting for growth or regulated by it. We used a curated list of macromolecular complexes spanning most cellular processes and calculated the proportion of each complex subunit that was growth-rate-dependent in each category (Figs 4A and S13 and Table S13) (Gene Ontology Consortium, 2019; Lock et al, 2019). As observed in prokaryotes and budding yeast, the top four categories relying the most on R-proteins belonged to a single process: the synthesis of proteins (Fig 4AB). Strikingly, R complexes were found at every single step of protein synthesis: the transcription of rRNAs and tRNAs and their processing, assembly and post-translational modification of the ribosome, and initiation and termination of translation (Fig 4B). Interestingly, expression of the chromatin-modifying complexes NuA4 and Ino80 were part of the R-sector (Fig 4C), suggesting they may be involved in ribosome biogenesis in fission yeast as has been proposed for NuA4 in budding yeast (Uprety et al, 2015). Alternatively, these results could indicate that the chromatin structure and levels of histone modification may be limiting for growth.

The overall correlation between growth and the factors involved in protein synthesis had a notable exception. Although RNA polymerase (RNAP) I and specific subunits of RNAPIII were part of the

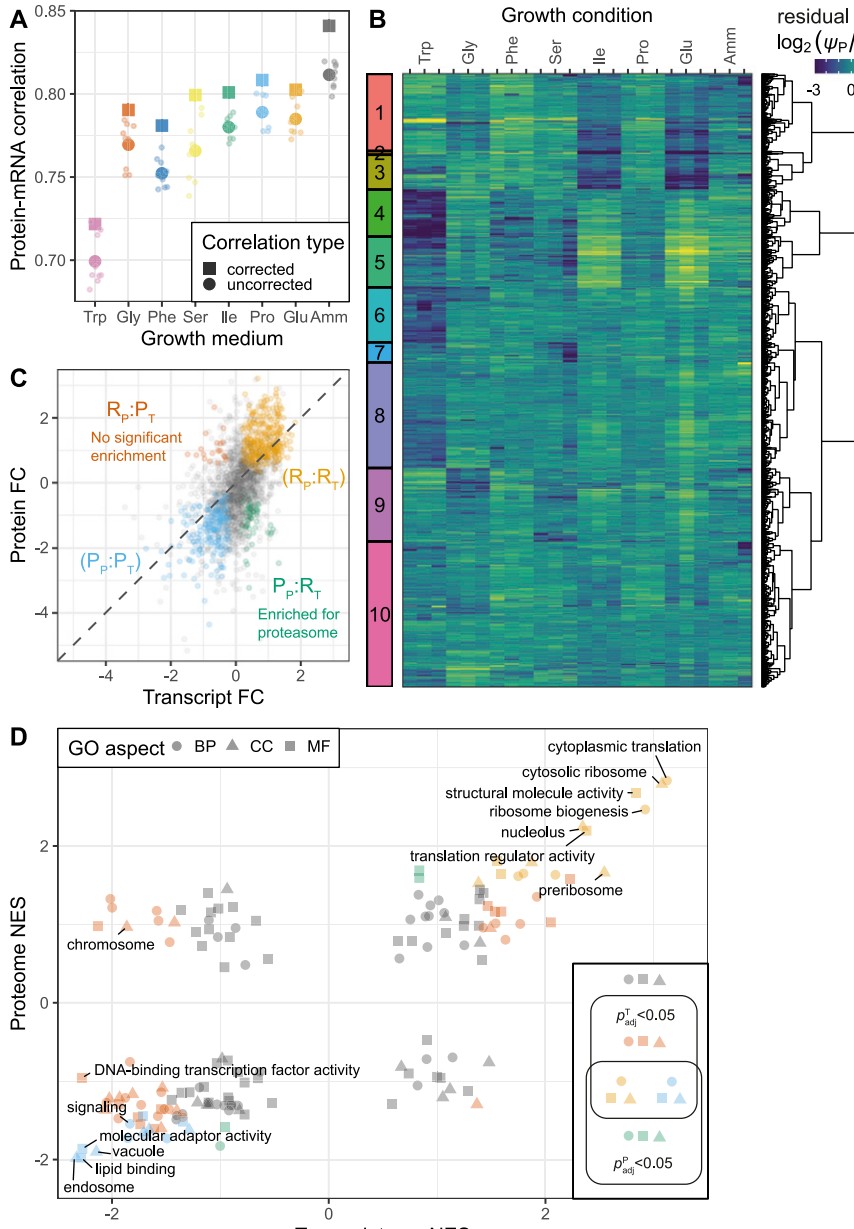

**Figure 3. Comparison of proteome and transcriptome.**
**(A)** Raw pairwise correlations between mRNA and protein number fractions (small circles) with their geometric means (large circles) and Spearman-corrected best estimates (squares). **(B)** Heat map of residual log₂-transformed ratios of protein and transcript number fractions ($\psi_P/\psi_M$) showing medium-specific differences in protein-to-mRNA ratio (see the Materials and Methods section). **(C)** Transcript and protein fold change (FC) measures of the repeated-median linear models (RMLMs) of 2,030 genes detected in both data sets. Colours indicate assignment to proteome and transcriptome sectors ($P_P$, $R_P$: protein assigned to P- and R-sector; $P_T$, $R_T$: transcripts assigned to P- and R-sector). An enrichment analysis of genes with non-matching protein and transcript sector assignments is shown in Fig S11. **(D)** Scatter plot of normalised enrichment scores (NES) of GO-slim–based gene set enrichment analyses in transcriptome and proteome. Genes were ranked according to the $P$-values associated with the transcript and protein RMLM, respectively. GO-slim terms are coloured as indicated according to the adjusted $P$-values from the gene set enrichment analyses, and terms with at least one $p_{adj} < 10^{-9}$ are labelled.

R-sector, RNA polymerase II specific subunits were not significantly correlated with growth rate (Fig 4B–D). Therefore, the number of RNAP II complexes is unlikely to be a limiting step in protein production during growth. Interestingly, RNAP II numbers were found to be limiting for the scaling of gene expression to cell size, indicating that coordination of gene expression to cell size and growth rate follow different mechanisms (Padovan-Merhar et al, 2015; Sun et al, 2020).

## The stoichiometry of translation complexes changes with the growth rate

Differences in FC values between protein complexes indicate that their relative levels or stoichiometry changes with the growth rate.

We hypothesised that these variations could provide mechanistic insights into the functioning of these complexes. To investigate this in the context of protein translation, we analysed three non-overlapping subclasses of translation proteins: the RP, the ribosome biogenesis regulon (RiBi), and the translation initiation, elongation and termination factors (IET) (see the Materials and Methods section, Table S14). The FC value for the IET class was the smallest of the three, whereas the trend line for RPs was the steepest (Figs 5A and S14A). As a result, the ratios between IET and RPs were significantly higher at slow growth (Fig 5B). It has been shown that RPs are held in reserve at slow growth rates (Metzl-Raz et al, 2017); these results suggest that an even larger fraction of IET and possibly RiBi proteins could be held in reserve. The relative

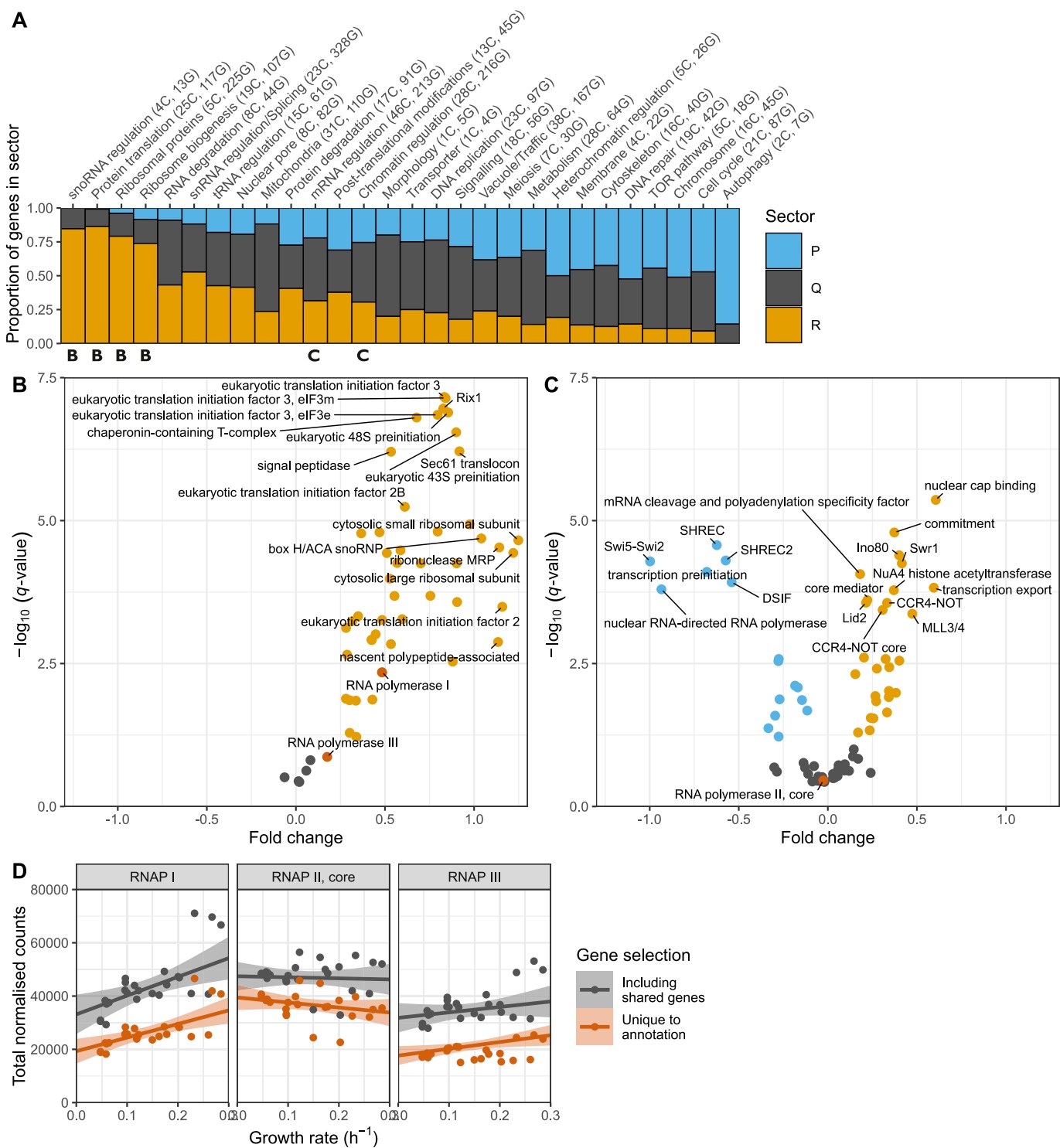

**Figure 4. Proteins from the R-sector are involved in every level of the protein production programme.**
**(A)** Fraction of R- (orange), P- (blue), and Q- (grey) proteins in manually curated broad categories of protein complexes. The number of complexes (C) and genes (G) in each category are shown in parentheses. The four leftmost categories encompass the protein production programme. **(B)** Volcano plot of protein complexes belonging to the broad categories "snoRNA regulation," "Protein translation," "Ribosomal proteins," and "Ribosome biogenesis" in the protein production programme. The plot shows the -log$_{10}$ of the $q$-value of the repeated-median linear model (RMLM) fit on the sum of normalised counts in each protein complex as a function of the growth rate against a normalised estimate of the slope of the fit (see the Materials and Methods section). **(C)** As shown in (B) for complexes belonging to the "mRNA regulation" and "Chromatin regulation" categories. **(D)** Sums of DESeq2-normalised counts for subunits of RNAP I (left), II (middle) and III (right) are plotted as a function of the growth rate. The sums of subunits unique to a given complex are plotted in orange and of all subunits are plotted in grey. RMLM fits are shown as lines and the predicted 2.5$^{th}$–97.5$^{th}$ percentile confidence interval (CI) of the fit as shaded areas.

abundances in EMM of IET:RiBi:RP were ~4:1:8 for the proteome mass fractions and 5:4:64 for the transcriptome RPKMs (Fig 5A and B). This confirms earlier observations that the burden on transcription for RP synthesis is higher than for the rest of the proteome (Schmidt et al, 2007; Marguerat et al, 2012). The growth laws for the initiation and elongation factors were almost identical to each other, suggesting constant stoichiometry with the growth rate (Fig S14B and C). Within the IET category, elongation factors were about three times as abundant as initiation factors, and about 50 times compared with termination factors (Fig S14B and C). This is in line with biochemical evidence showing that translation initiation is a limiting step for protein synthesis (Aylett & Ban, 2017). Taken together, we have shown how the growth law can inform on the regulation of gene expression through changes in the stoichiometry of factors with the growth rate.

Furthermore, the large burden of RPs during fast growth resulted from the coordinated growth-related expression of most individual RPs and from a growth dependence component steeper than that of IET and RiBi (Fig 5C). This indicates that the aggregate burden of RPs results from coordinated regulation at the level of single genes (Petibon et al, 2020). The IET and RiBi categories also contained more proteins that were assigned to the P- and Q-sectors, and whose expression data were not well explained by the robust model because of significant condition-dependent expression (Figs S15 and S16). For instance, the initiation factor eIF3e was present in sub-stoichiometric amounts relative to eIF3a. Interestingly eIF3e has been shown to selectively regulate the translation of transcripts coding for metabolic enzymes (Shah et al, 2016).

16 protein groups annotated as RPs were assigned to the Q-sector because their expression was not significantly positively correlated with the growth rate and we here explore these Q-RPs further (Fig S16). Their relative protein abundances were slightly lower than those of RPs that did belong to the R sector (R-RPs, see Fig S16A). However, median transcript abundances and FC values were not significantly different between Q-RPs and R-RPs (Fig S16B). This opens the possibility of regulatory feedback at the post-transcriptional level. Interestingly, half of these Q-RPs (Rlp7, Rpl102, Rpl2501, Rpl35A02, Rps1502, Rps20, Rps27, Rps2801, and Rps2802) are annotated with functions in ribosome biogenesis on the PomBase database (Lock et al, 2019). In addition, the budding yeast orthologue of Q-RP Rps20 has been proposed to regulate RNAPIII transcription, providing a potential link between ribosomes and tRNA synthesis (Warner & McIntosh, 2009). The proteome expression data for all Q-RPs is plotted in Fig S16C. Together, this suggests that Q-RPs could be attractive candidate proteins that could have additional functions outside of the ribosome.

Principles of proteome allocation are often conserved in prokaryotes and eukaryotes despite significant mechanistic differences in the way genes are transcribed and translated (Dai & Zhu, 2020). Therefore, we thought to compare our findings in fission yeast with published data sets from the budding yeast *S. cerevisiae* and the bacterium *E. coli* (Schmidt et al, 2016; Metzl-Raz et al, 2017). We reanalysed published proteomics data for *E. coli* cells growing at different rates in a series of environmental conditions to extract the relative proteome fractions, and we subsequently computed the growth law parameters for translational proteins (see the Materials and Methods section, Table S15) (Schmidt et al, 2016). For

*S. cerevisiae*, we merely used growth law parameters of RPs published elsewhere (Metzl-Raz et al, 2017). We found that *E. coli* could sustain a given growth rate with a smaller fraction of RPs than both yeasts, which was due to a smaller growth law slope (Fig 5D). This suggests that the effective translation rate in the yeasts is lower than that of *E. coli*. Among the two yeasts, fission yeast used its RPs significantly more efficiently—using a smaller RP mass fraction to sustain any given growth rate than the budding yeast trend line—but the effect could not be assigned to a significant difference in either the slope or the intercept parameter specifically. Next, we asked whether the changes in stoichiometry of translational proteins during slow growth were conserved in *E. coli*. Again, both the IET/RP and RiBi/RP ratios were higher during slower growth (Fig S17A and B) because the individual RPs had steeper growth laws (Fig S17C). A steeper growth law of RPs than that of elongation factors was recently predicted by a model of *E. coli* that minimised the total expression cost (Hu et al, 2020). Our results indicate that allocation strategies are conserved even though protein production differs mechanistically between the two kingdoms.

## P-sector proteins are part of the core environmental stress response programme

To complement our analysis of the R-sector, we next examined fission yeast proteins from the P-sector, that is, proteins with a negative growth-dependent component. In contrast to the R-sector clusters 1 and 2, we could not identify P-sector clusters whose expression could be explained exclusively by a negative growth rate correlation (Fig 2A–C). This indicates that proteins with a strong P component are also often regulated in response to specific nitrogen sources. Moreover, the growth component for P-proteins was less significant overall than for R-proteins (Fig S18A and B). These results suggest that regulation of the R- and P-sectors may differ mechanistically.

Unlike R-proteins, which are mostly involved in protein production, P-proteins belonged to a diverse set of complexes participating in a large array of functions (Fig 4A). As individual proteins, they showed weaker correlations than R sector complexes (Fig S18C and D). To analyse whether this diverse set of P-proteins was participating in a common higher level functional programme we analysed the fission yeast GO-slims alongside 21 lists covering fission yeast physiology and environmental responses (Figs 5E and S19A) (Mata et al, 2002; Chen et al, 2003; Rustici et al, 2004; Marguerat et al, 2012; Rallis et al, 2013; Saint et al, 2019; Kamrad et al, 2020). Functional classes with strong P-sector components included vacuole biology, endosome and phagosome, transport and genes induced in the adaptation to nitrogen removal, and/or after treatment with caffeine and rapamycin. The latter two classes, which had the strongest response, are thought to be controlled by TORC1 (Mata et al, 2002; Rallis et al, 2013). This suggests that nitrogen sources supporting slower growth rates trigger a form of metabolic stress response. Accordingly, the total expression of the fission yeast core environmental stress response programme up-regulated genes (CESR up) was negatively correlated with the growth rate (Fig 5F). This stress module comprises genes induced in response to a wide range of environmental and genetic perturbations (Chen et al, 2003; Pancaldi et al, 2010). Conversely, genes down-regulated as part of the

**Figure 5. Stoichiometries of translation complexes, comparison of ribosomal growth law with other species, and functional analysis of P-sector.**
**(A)** Sum of the protein fractions plotted as a function of the growth rate for factors involved in translation initiation, elongation, and termination (IET; left), ribosome biogenesis (RiBi; middle), or ribosomal proteins (RP; right). The best fit and bootstrapped 95% confidence interval (CI) are shown in black and grey, respectively. The fold change (FC) values ± standard deviations of the bootstrapped values are shown. **(B)** Proteome mass ratio plotted as a function of the growth rate for the following comparisons: IET versus RP (left), RiBi versus IET (middle), and RiBi versus RP (right). Shown in black/grey are the predictions and 95% CIs as given by the linear models fitted to the data in (A). **(C)** FC values for proteins of the IET, RiBi, and RP categories plotted as a function of their median expression. Proteins assigned to the R-, P-, and Q-sectors are coloured in orange, blue, and grey, respectively. **(D)** Total proteome mass fraction allocated to RPs as a function of growth rate for *S. cerevisiae* (red) (Metzl-Raz et al, 2017), *Schizosaccharomyces pombe* (green), and *Escherichia coli* (grey) (data from Schmidt et al [2016]). RMLM fits and 95% CIs are shown as lines and shaded areas, respectively. **(E)** The −log₁₀ Q-value of repeated-median linear model (RMLM) fits plotted against their respective FC values for proteins belonging to GO-slim and

CESR response (CESR down, also called growth module) belonged to the R-sector (Figs 5F and G and S19B). This finding validates the longstanding hypothesis that the balanced expression of the fission yeast stress response is quantitatively connected with the growth rate (López-Maury et al, 2008). In addition, P-proteins were enriched for factors regulated during the S phases of the cell cycle, which is consistent with evidence that the cell-cycle phase length differs between nitrogen sources, in particular growth on Trp (Figs 5E and S19C and D) (Carlson et al, 1999; Rustici et al, 2004). Notably, we did not observe a simple relationship between the expression of cell cycle markers and the growth rate (Fig S19E). This is in line with earlier flow cytometry and microscopy data, which did not find a straightforward relationship between the length of cell cycle phases and the growth rate upon growth on different nitrogen sources (Carlson et al, 1999).

Notably, the functional classes involved in metabolism were not strongly negatively correlated with the growth rate (Fig 5E), and the fission yeast P-sector was only marginally enriched in proteins involved in central and energy metabolism (Fig S20). This contrasts with previous data from *E. coli* and *S. cerevisiae* where metabolic genes have been reported to be important components of the P-sector (Hui et al, 2015; Schmidt et al, 2016; Metzl-Raz et al, 2017). However, when considered globally, the sum of protein mass fractions dedicated to metabolic enzymes was clearly anti-correlated with growth in fission yeast, ranging from ~70% of the proteome in poor nitrogen sources to ~55% in the fastest media (Fig 6A). This indicates that in our system which does not rely on titration of a limiting nutrient to modulate the growth rate, the total protein burden on metabolism is linked to the growth rate, whereas allocation to specific enzymes is not. Therefore, the global anti-correlation of metabolic enzymes with growth rate observed in our data may be a manifestation of the trade-off between metabolism and translation, and not the result of the direct quantitative regulation of metabolic enzymes expression with the growth rate.

### The burden of specific metabolic pathways is principally condition-dependent

On top of the growth-dependent components, many fission yeast proteins show clear condition-specific gene regulation (Fig 2A–C). Functional analysis indicated an enrichment of these genes for functions related to metabolism. This is consistent with the adoption of distinct metabolic allocation strategies in response to growth with different nitrogen sources (Alam et al, 2016; Mülleder et al, 2016). We classified metabolic genes into six non-overlapping classes based on the following GO terms: canonical glycolysis (GO:0061621), generation of precursors and energy (GO:0006091), cellular amino acid metabolic process (GO:0006520, which includes the interconversion of ammonium, glutamate, and glutamine), lipid metabolic process (GO:0006629), vitamin metabolic process (GO:0006766), and all other metabolic pathways (including transport of

metabolites) (Figs 6B and S21 and Table S16). To avoid over-estimating the burden of gene expression by double-counting genes assigned to multiple terms, each protein was assigned only to the first of these GO-terms it was annotated with. The relative allocation to each class was condition-specific, indicating that metabolic states rely differentially on specific pathways (Fig 6B). We note that similar growth rates can be supported by different allocation strategies, as in the case of the Trp and Gly containing media in which cells channelled resources preferentially towards glycolysis (Trp) or amino acid metabolism (Gly) (Figs 6B and S21). The growth-related components of those categories were weak, except for the vitamin metabolism proteins which belonged to the R-sector and the precursor/energy proteins that showed a significant P component (see below, Fig S21). Most coenzymes are stable molecules synthesised only as much as necessary to support growth (Hartl et al, 2017). The strong positive correlation of vitamin metabolism expression with growth rate suggests that cells also minimise the translation burden of vitamin metabolic enzymes. In summary, expression of metabolic enzymes in our system, although connected to the growth rate, is mainly condition- and pathway-specific.

We next took a closer look at the energy metabolism pathways and their negative correlation with the growth rate. Nutrient quality, cell growth, and energy metabolism are intimately connected. The generation of ATP through fermentation is often favoured in conditions that support faster growth, whereas slow-growing cells in limiting conditions tend to switch to respiratory metabolism (Vander Heiden et al, 2009; Shimizu & Matsuoka, 2018). Therefore, we asked whether protein allocation to either energy metabolism pathway was correlated with the nitrogen sources used and/or growth rate. To this end, we split the non-glycolytic generation of precursors and energy category into the fermentative enzymes pyruvate decarboxylase (Pdc101) and alcohol dehydrogenase (Adh1), and the respiration process into tricarboxylic acid cycle (TCA, GO: 0006099) and oxidative phosphorylation (OXPHOS, GO:0006119) enzymes (Fig 6C and S22). Surprisingly, none of the categories were consistently correlated with the growth rate. Instead, condition-specific expression was dominant, and a clear repression of all OXPHOS complexes upon growth on serine was observed (Fig S23). A recent report showed that serine catabolism generates high levels of reactive oxygen species (ROS) in *S. pombe*, suggesting that respiration may be repressed upon growth on serine to avoid a further increase in ROS (Kanou et al, 2020). Notably, expression of the fermentative enzymes Adh1 and Pdc101, although variable between conditions, was consistently higher than the total expression of the respiratory enzymes. Moreover, respiratory enzymes were not induced in nitrogen sources supporting slow growth. Taken together, the expression balance between fermentation and respiratory enzymes was not quantitatively connected to the growth rate, but depended on the nutrient properties.

literature lists (Mata et al, 2002; Chen et al, 2003; Rustici et al, 2004; Rallis et al, 2013; Kamrad et al, 2020). List with a significant negative slope ($q$-value < 0.001) are highlighted in blue. BP GO-slim terms related to metabolism are highlighted in green, stress/growth modules from Chen et al (2003) in vermillion, and cell cycle induced modules from Rustici et al (2004) in orange. **(F)** Sum of protein fractions plotted as a function of the growth rate for the core environmental stress response (CESR) repressed (growth module) or induced (stress module) genes. RMLM fit and predicted 95% CI as in (A). **(G)** Assignment of growth and stress module proteins (Prot) detected in all samples and their respective transcripts (Trans) to the R- (orange), P- (blue), and Q- (grey) sectors based on protein fraction expression and DESeq2 normalised counts. Each protein is connected to its corresponding transcript by a line and the colours correspond to the protein sectors.

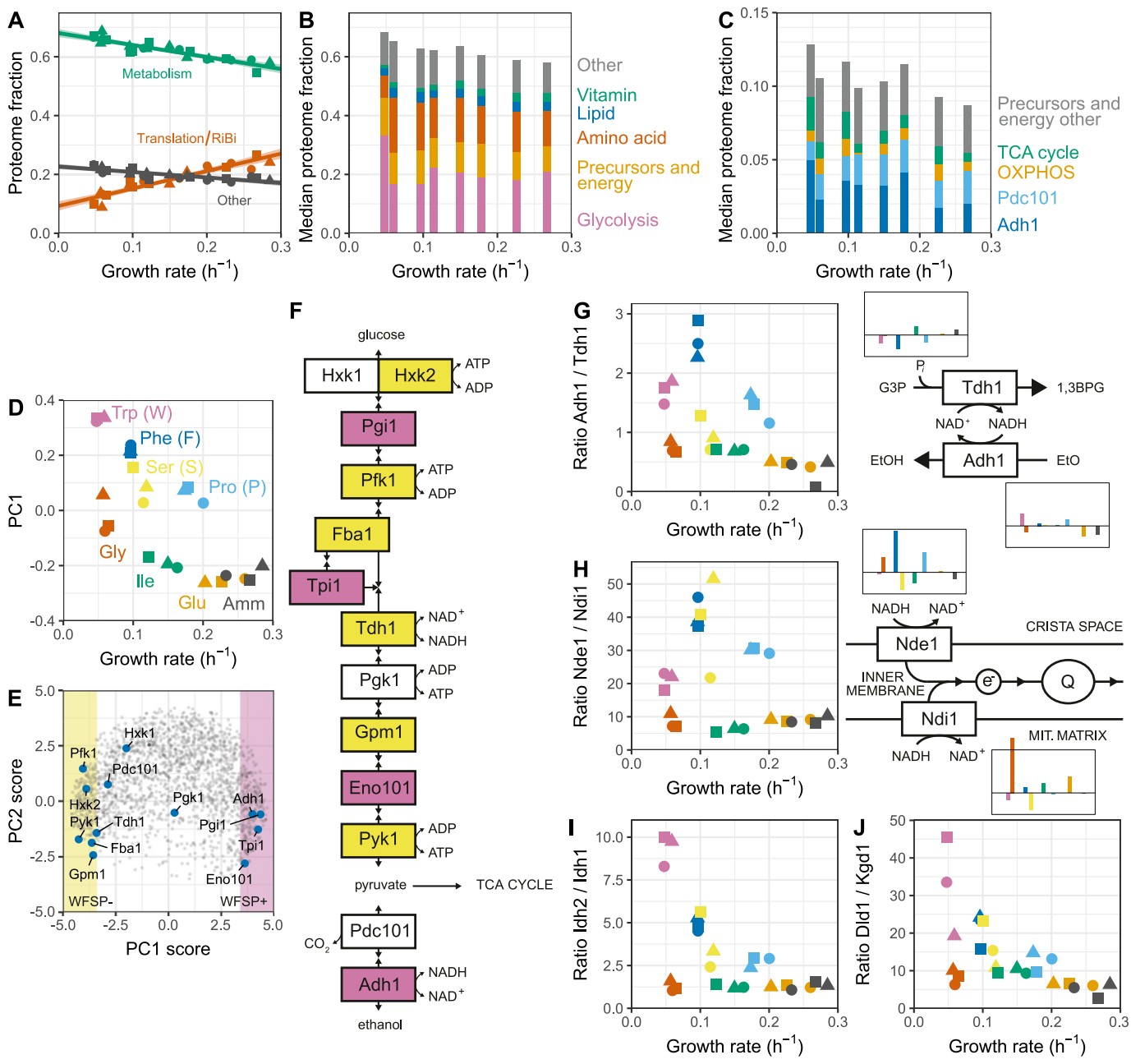

**Figure 6. The coordination of energy metabolism enzymes with the growth rate is marginal.**
**(A)** Sum of protein fractions of proteins involved in translation and ribosome biogenesis (red, see Fig 5A), energy metabolism, and transport (green, see Fig 5E) or all other genes (grey) plotted as a function of the growth rate. **(B)** Relative proteome fractions of five categories of proteins involved in metabolism. The median of the three replicates from each condition was used for calculating the protein fractions and plotting growth rates. **(C)** As shown in (B), for proteins of the OXPHOS and TCA pathways, the Adh1 and Pdc101 fermentation proteins, and proteins annotated as "generation of precursor metabolites and energy" and not included in the other four categories or glycolysis. **(D)** Protein expression as a function of growth rate as exhibited by the first principal component (PC1). **(E)** Comparison of the first two principal components (principal component analysis biplot) for each protein group detected in the proteome across all conditions. Areas with >50% variance explained by PC1 correlation are highlighted in yellow (negative correlation, WFSP–) and pink (positive correlation, WFSP+). Genes related to glycolysis and ethanol fermentation are indicated in blue. **(F)** Topology of the glycolysis and ethanol fermentation pathway showing genes, cofactors, and selected metabolites, with colours as in (E). **(G)** Left: ratio of protein fractions for Adh1/Tdh1 plotted as a function of the growth rate. Right: diagram showing Adh1 and Tdh1 functioning together with median proteome fractions of both proteins in each condition. Colours are as annotated in (D). **(H)** As shown in G for Nde1 and Ndi1. **(I)** Ratio of protein fractions of Idh2 and Idh1 plotted as a function of the growth rate. Colours are as annotated in (D). **(J)** As shown in I for the ratio of the protein fractions of Dld1 and Kgd1 plotted as a function of growth rate.

To complement this analysis, we searched for condition-specific patterns of protein expression that were not related to the growth rate in our proteomics data set using principal component analysis (Fig S24A–E). The first principal component (PC1) explained 29% of the total variance and split the culture conditions in two irrespective of the growth rate with Trp (W), Phe (F), Ser (S), and Pro (P)

in one group (from here on termed the WFSP media) and Gly (G), Ile (I), Glu (E), and Amm in the other (Fig 6D). Strikingly, 24% (474/1,988) of proteins had more than 50% of their variance explained by PC1. We defined two large classes of protein based on their response to this component: (i) WFSP+ consisting of 259 proteins that were positively correlated with PC1 and therefore induced in the WFSP media; (ii) WFSP– characterised by 215 proteins with expression negatively correlated with PC1 and therefore repressed in the WFSP media (Table S17). Interestingly, no single principal component was dominated by growth rate correlation (Fig S24E), reinforcing the point that nutrient-specific and growth-dependent components of gene expression coexist for many proteins.

Glycolytic and NAD-dependent enzymes were the two major classes of proteins overrepresented in the WFSP lists. First, most glycolytic enzymes belonged to one of the two WFSP classes (Figs 6E and F and S25). These enzymes were highly expressed across conditions, amounting to ~15–30% of the total proteome mass (Figs 6B and S21). Therefore, the total gene expression burden of cellular metabolism across the WFSP conditions was heavily affected by the abundance of a small number of enzymes. Second, the two enzymes glyceraldehyde-3-phosphate (G3P) dehydrogenase Tdh1 and alcohol dehydrogenase Adh1 were assigned to opposing WFSP lists, and the ratio of Adh1/Tdh1 protein abundance was highly elevated in the WFSP conditions (Fig 6F and G and S26). Fermentation of a single molecule of glucose generates two molecules of ethanol and carbon dioxide. During the process, Tdh1 reduces two NAD+ molecules and Adh1 oxidises two NADH molecules. Therefore, the elevated Adh1/Tdh1 balance exerts a pressure on the NAD+/NADH equilibrium towards the NAD+ side. The induction of Adh1 and repression of Tdh1 proteins may be a controlled response to maintain homeostasis under disruptions to the NAD+/NADH redox balance. This way, differential resource allocation towards the NAD-cycling glycolytic fermentation pathway may indicate that the metabolic rewiring invoked by the WFSP nitrogen sources could result from changes in the cell redox balance.

To follow up on this observation, we further investigated the burden of NAD-dependent pathways. NADH is oxidised by NADH dehydrogenases that are situated in the inner mitochondrial membrane; the enzyme transfers two electrons per NADH molecule to the electron transport chain to power ATP synthesis. On the other hand, NAD+ is reduced several times during each iteration of the TCA cycle by the α-ketoglutarate (αKG) dehydrogenase complex (KGDHC), the isocitrate dehydrogenase (IDH) complex, and the malic enzymes. Fission yeast is thought to have two separate NADH dehydrogenase enzymes, Ndi1 and Nde1, with the NAD-binding domain of Ndi1 facing the mitochondrion and Nde1 facing the cytosol. We examined the expression burden of these enzymes in our data and found that, although neither belonged to one of the WFSP lists, the ratio of Nde1/Ndi1 expression was strongly elevated in the WFSP conditions (Figs 6H and S26). The IDH complex comprises the two subunits Idh1 and Idh2, and KGDHC consists of four subunits: Kgd1, Kgd2, Ymr31, and Dld1, the latter being part of multiple complexes. Dld1 and Idh2 were part of the WFSP+ class, unlike any of the other subunits. As above, the ratio of protein abundances for Dld1/Kgd1 and Idh2/Idh1 were elevated in the WFSP conditions (Figs 6I and J and S26). Therefore, the response to the

WFSP nitrogen sources altered the stoichiometry of NAD-dependent enzymatic complexes.

Importantly, these signatures were not detected in our transcriptomics data, suggesting a role for post-transcriptional regulation. In line with this, ubiquitin and its related pathways, as well as the translation factors eIF3e and eIF5a, showed strong WFSP patterns suggesting a role for protein stability (Figs S4E and S15B–D and Table S17). In summary, we identified two distinct cellular states that differed in the expression of enzymes involved in fermentation and the cell's redox balance that were not correlated with the growth rate.

### Correcting for growth rate dependence revealed additional transcriptional signatures of growth on single amino acid sources

Defining the heterogeneity of metabolic states is key to a mechanistic understanding of cell population evolution, but this requires disentangling the gene signatures that depend on the growth rate from those that are purely nutrient specific. Our data set has the unique capacity to achieve this. We performed differential expression analysis on our RNA-Seq data set, by comparing each growth condition to a reference transcriptome obtained via averaging all the conditions, and corrected for the growth-dependent component of gene expression (see the Materials and Methods section). We defined 10 signatures (termed R1–R10) by clustering the log₂-transformed fold change ratios with respect to the synthetic reference of all genes that were significantly enriched in at least one condition (Figs 7A and S27 and Table S18).

The 10 signatures covered the differential expression of 2,140 genes in total, representing ~43% of the fission yeast transcriptome. Five signatures (R2, R3, R5, R6, and R8) were also visible at the proteome level (Fig 7B). About 67% of the mRNA present in the transcriptomic signatures were quantified in at least one condition in the proteome and ~38% were detected in all conditions, indicating that this relatively limited agreement was not due to the lower coverage of the proteomics data.

We next performed functional enrichment analyses of the transcriptomics clusters (see the Materials and Methods section), using Gene Ontology annotations (Gene Ontology Consortium, 2019; Lock et al, 2019). Broader functional categories were captured using GO-slim analysis (Fig 7C), and specific pathways using terms from the biological_process ontology with at most 50 annotations. List overlap analyses (Fig S28) as well as gene set enrichment analyses, ranking genes based on their log₂ fold change over the synthetic reference after shrinkage, were performed for each growth medium (Table S19 and Fig S29). In agreement with our observation that respiratory genes were repressed in Ser medium, the Ser repressed cluster R3 was strongly enriched for genes related to mitochondrial metabolism. In addition, genes from clusters R6 and R10, which were induced in the Ser medium, were enriched for detoxification (Figs S28 and S29). The Ser response also contained oxidoreductases and proteins involved in metal ion homeostasis, which is compatible with the recently reported high levels of ROS generated by serine catabolism (Kanou et al, 2020). The Trp repressed cluster R2 was enriched for genes related to amino acid metabolism (Fig S28) and the corresponding GO-term also had a negative NES value (Fig S29), again suggesting that the slow growth sustained by the Trp

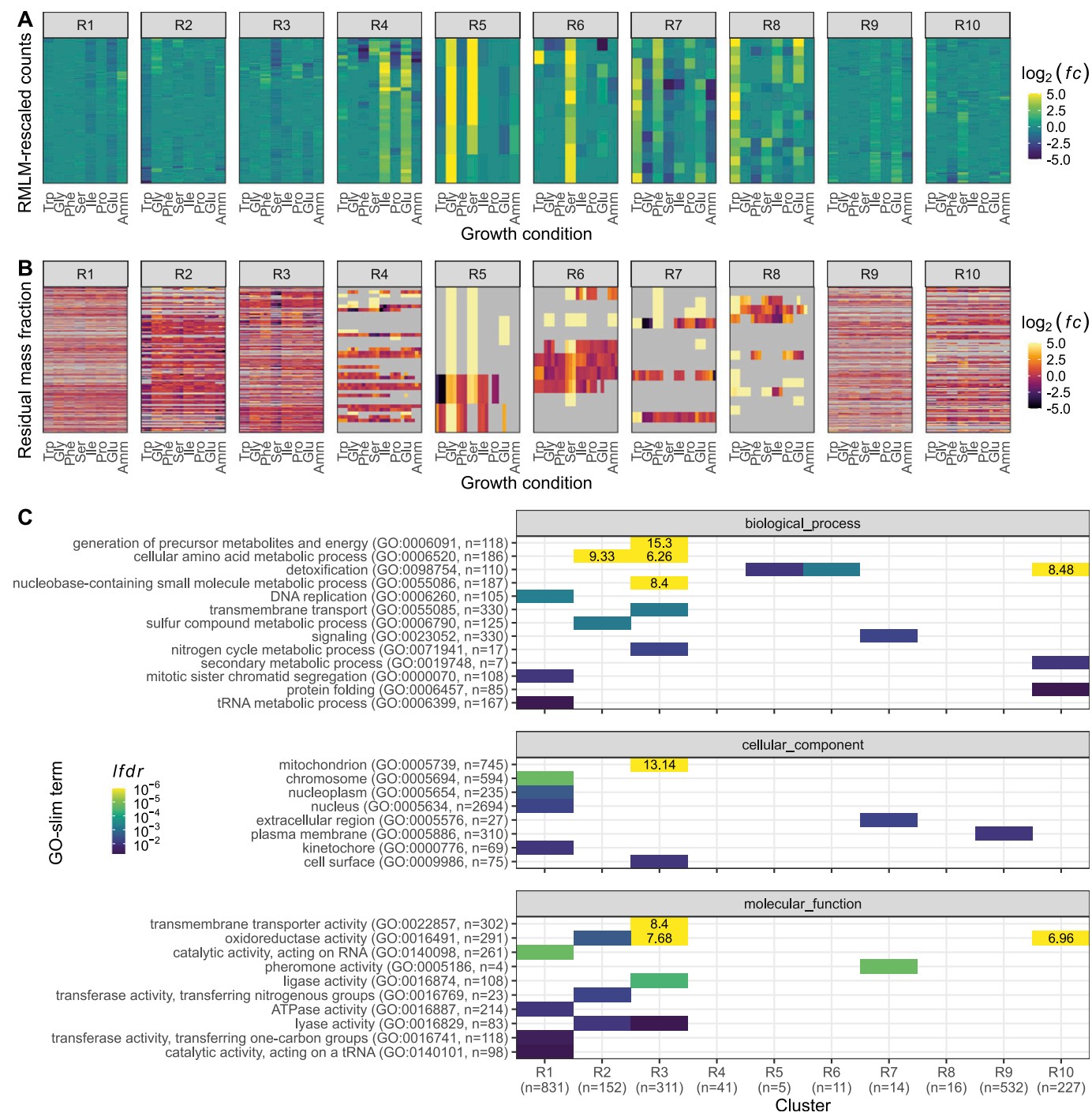

**Figure 7. Transcriptomic signatures for growth on amino acid sources.**
**(A)** DESeq2 log₂ fold change ratios after shrinkage for the 10 signatures R1-R10 (scale capped at abs(log₂(fc)) = 5). Fold changes are relative to the RMLM-predicted synthetic reference (see the Materials and Methods section). Columns are ordered according to the growth rate and rows are ordered by hierarchical clustering ([Fig S27]). **(B)** The log₂-transformed ratios of observed versus RMLM-predicted protein fractions for genes in the R1-R10 signatures. Row and column orders are as described in (A). Genes missing from the proteomics data are in grey. **(C)** Functional analysis of the transcriptomics clusters R1-R10 as shown in (A). Enrichment for GO slim terms belonging to the "biological process" (top), "cellular component" (middle), and "molecular function" (bottom) categories are shown. The colour scheme denotes the local false discovery rate (*lfdr*, capped at 1 × 10⁻⁶ and printed on the figure if capped) from a Fisher's exact one-sided test for the overlap of each cluster with functional lists. Only significant lists are shown (*lfdr* < 0.05) and the number of genes in each category and cluster are shown in parentheses.

medium was not due to any additional expression burden of disrupted amino acid synthesis. The small cluster R7 was enriched for genes related to pheromone activity (M-factor precursors), signalling, and the induction of meiosis (Fig S28). Interestingly, the signature expression across conditions for these genes (induced in Trp, Phe, Pro, and Glu containing media) mirrored that of Mae2 (Fig S4F), which removes excess carbon from the TCA cycle. As meiosis is usually induced by nitrogen starvation (Petersen & Russell, 2016), this result suggests that the state of CCM may also play a role in the meiotic transition, as (elemental) nitrogen was abundant in all growth media used. Altogether, we identified a rich set of metabolic signatures that were not dependent on the growth rate, but exclusively reflect changes in external nutrients.

## Discussion

In this study, we quantified the proteome and transcriptome of the fission yeast *S. pombe* grown in eight defined media that affect the growth rate. Each medium contained a single nonlimiting source of nitrogen, such that variations in gene expression were determined by system-level resource allocation and not by the response of a single pathway to the titration of a limiting nutrient. This setup is in contrast to other studies which relied on a specific limiting nutrient to perturb resource allocation while affecting the growth rate (Brauer et al, 2008; Hui et al, 2015), or leaving it constant (Yu et al, 2020). In chemostats, the growth rate is affected externally by the nutrient quantity, and the same growth rate can be obtained by limiting several different nutrients (Airoldi et al, 2016). Our turbidostat cultures were more alike to continuous flask cultures, where the growth rate was determined by internal allocation in response to the nutrient quality. This made it possible to determine the effect on growth and metabolism of the medium composition because each growth medium gave rise to a reproducible steady state.

Using this orthogonal approach, we propose a model in which shifts in resource allocation trigger two layers of gene expression regulation. The first layer consists of gene expression that is significantly correlated with growth rate and the second is condition specific, depending solely on nutrients. Many proteins and mRNAs showed a combination of both layers of regulation. This suggests that condition-specific responses occur on top of a global level gene regulation that is coordinated with the growth rate (Shahrezaei & Marguerat, 2015). Importantly, the global layer of regulation discussed here affects relative abundances of proteins and of mRNAs, and we did not obtain data on the size of the total mRNA and protein pools. It is therefore distinct from the scaling of gene expression to the growth rate which ensures constant biomolecule concentrations (Chávez et al, 2016). The mechanisms behind the observation that a large number of mRNA and proteins show some level of global scaling with the growth rate are not entirely clear. It could be related to the fact that expression of the protein production machinery itself increases with the growth rate and to changes in levels of TOR signalling for instance (see below). This could result in different cellular states that feedback globally on gene expression (Keren et al, 2013). It is of note that the growth-rate-dependent component defined in this study might in some cases complicate the interpretation of

condition-specific responses and should then be taken into account (Pancaldi et al, 2010; Yu et al, 2021).

Eukaryotic growth-rate-related gene expression depends to some extent on the TORC1 axis of gene regulation, which is widely conserved across eukaryotes (Weisman, 2016; González & Hall, 2017; Morozumi & Shiozaki, 2021). TORC1 activity is affected by a variety of stressors including nutrient starvation. Upstream of TORC1, the adenosine monophosphate kinase AMPK has been proposed to mediate the response to nitrogen starvation, and intriguingly, the two complexes can inhibit each other (Davie et al, 2015; Ling et al, 2020). Downstream, the TORC1 pathway is a key regulator of the balance between the stress and growth modules (López-Maury et al, 2008; Rallis et al, 2013, 2014), with targets including eukaryotic initiation factor 2 subunit $\alpha$ (eIF2$\alpha$) (Valbuena et al, 2012), the SAGA complex (Laboucarié et al, 2017), and the rate of fermentation through Greatwall and PP2A$^{B55\delta}$ (Watanabe et al, 2019). These questions are often studied during adaptation to changing conditions and our system using continuous culture in turbidostats provides an attractive setup for future studies of the mechanisms that maintain the stress versus growth gene expression balance in steady-state conditions.

We found that known chromatin modifiers belonged to the R-sector. This is intriguing as expression of histones themselves was not dependent on the growth rate (Table S6). This may suggest that number of histones modifying enzymes and levels of modifications are rate limiting for transcription, or alternatively mediate an orthogonal function such as signalling the cell metabolic state through covalent protein modifications (Mellor, 2016; Figlia et al, 2020; Morgan & Shilatifard, 2020). This illustrates the intricate relationship of chromatin structure with the cell metabolism. Moreover, we found that RNAP II expression was not increasing with the growth rate suggesting that, unlike for gene expression scaling to cell size, its numbers are not limiting for the rate of growth (Padovan-Merhar et al, 2015; Sun et al, 2020). Yet, maintaining constant mRNA concentrations requires synthesis or degradation rates to adjust to cell growth. Therefore, other mechanisms such as transcription elongation or mRNA decay rates are likely to be modulated with the growth rate as suggested in budding yeast (Chávez et al, 2016).

Discussing protein allocation in term of factors limiting for growth relies on the assumption that expression of all proteins is optimised for growth in any given condition. Recent evidence has challenged this view and has suggested that significant parts of *E. coli* (Valgepea et al, 2013; Peebo et al, 2015; Mori et al, 2017) and budding yeast (Metzl-Raz et al, 2017; Yu et al, 2020) gene expression are not immediately required for sustaining the growth rate and are instead held in reserve. This reserve pool of protein could support cell adaption to sudden environmental changes. It has furthermore been suggested that CCM has a large reserve capacity, suggesting that many enzymes may also not be used solely to maximise metabolic fluxes (O'Brien et al, 2016; Christodoulou et al, 2018; Yu et al, 2020). In this study, whereas several nutrient-specific regulatory programmes were detected in both the transcriptome and the proteome, such as specific responses to Ser and Trp, this was not true for the WFSP pattern and other transcriptomics signatures (Figs 6 and 7). This disconnect could means that metabolic pathways are differentially buffered through protein levels and stability

which could in turn be interpreted in term of reserve capacity. A better understanding of post-transcriptional regulation in fission yeast will be important to fully understand what causes the high translational burden of metabolism.

We found that expression of metabolic enzymes was strongly condition-specific and only marginally anti-correlated with the growth rate. This condition-specific regulation represented a large change in the gene expression burden, driven by glycolytic proteins and enzymes and complexes relying on NAD turnover. Interestingly, this large variation in expression burden of the carbon metabolism resulted from changes in nitrogen source and occurred in the presence of abundant external glucose. This highlights the fact that metabolic adaptation to external condition is pervasive not only in term of fluxes but also in term of gene expression burden. The catabolism of the backbones of the amino acids used as nitrogen sources could provide a link between nitrogen and carbon metabolism in our system. Our data provide a rich resource to constrain future genome-scale models of fission yeast that integrate metabolism and gene expression, which will allow testing this hypothesis (O'Brien et al, 2013; Sánchez et al, 2017; Chen et al, 2021).

An improved understanding of the fundamental principles behind cellular growth and the physiological and translational burden of metabolism across evolutionarily diverse biological systems would influence a wide range of research areas such as microbiology, synthetic biology, and cancer research. Cellular models of growth should integrate strategies used by a variety of organisms under a wide range of conditions, to identify common principles. Beyond its contribution to our understanding of gene regulation, this work will support future experimental and modelling efforts aimed at defining the nature of the trade-offs involved in growth, stress resistance, and metabolism across the tree of life.

# Materials and Methods

## Culture conditions

Cells were grown in continuous culture in turbidostats using Edinburgh minimal media (EMM2) with saturating amounts of carbon and nitrogen (Petersen & Russell, 2016). This ensured that the cells could reach balanced exponential growth, limited only by internal gene expression patterns. In addition to the standard EMM2 media where nitrogen is provided by 93.5 mM of ammonium chloride ($NH_4Cl$, referred to as Amm), we used seven alternative nitrogen sources where 20 mM of a single amino acid replaced the $NH_4Cl$: glutamate (Glu), proline (Pro), isoleucine (Ile), serine (Ser), phenylalanine (Phe), glycine (Gly), and tryptophan (Trp) (Sigma-Aldrich).

Cells were grown and harvested as follows: 972 h⁻ cells from frozen glycerol stocks were precultured on YES agar plates. Single colonies were inoculated in 5–10 ml of EMM2 in glass flasks and grown overnight at 32°C. Approximately 1 ml of culture was transferred to a fresh flask containing EMM2 and the final nitrogen source and grown to large ~$5 \times 10^6$ cells/ml. These cells were used to inoculate the continuous culture setup at 0.5–$1 \times 10^6$ cells/ml. The process was repeated for biological triplicates grown from three different colonies.

To generate the final cultures, cells were grown in turbidostats (Takahashi et al, 2015), with media flow controlled using customised

Python scripts (Saint et al, 2019). Cell cultures were monitored every 30 s and fresh growth medium was added whenever the optical density $OD_{600}$ exceeded 0.4. This resulted in 1–2% dilution cycles, keeping the total culture volume constant throughout. Cells were kept in the turbidostats for ~10 generations at 32°C. To measure the growth rate, cells were diluted twofold approximately halfway through the experiment and regrown to the reference level of $OD_{600}$ = 0.4. The growth rate for each sample was determined by fitting an exponential curve to the OD measures acquired every 30 s during the regrowth phase. The final culture volumes were ~30 ml, from which 10 ml was used for transcriptomics, 10 ml for proteomics analysis, and 10 ml was saved as a backup. The cells were harvested by centrifugation, washed twice with PBS and stored at –80°C until RNA-Seq and proteomics sample preparation was performed.

## RNA-Seq

A 10-ml aliquot of the culture was centrifuged at 3,000 rpm for 3 min in a 5810R Eppendorf centrifuge. After removing the supernatant, cell pellets were frozen in dry ice and kept at –80°C until the library preparation was performed. Total RNA from the pellets was extracted using the hot-phenol method (Lyne et al, 2003) and the RNA obtained was quantified using a BioDrop (biochrom). Poly(A) enrichment was performed using 500 ng of total RNA with the NEBNext Poly(A) mRNA Magnetic Isolation Module (NEB) kit according to the manufacturer's instructions. The remaining mRNA was used for stranded RNA-Seq library preparation using the NEBNext Ultra II Directional RNA Library Prep Kit for Illumina (NEB) according to the manufacturer's instructions. The resulting libraries were quality checked and quantified using the Bioanalyser (Agilent) and a Qubit dsDNA BR Assay Kit (Invitrogen), respectively.

Libraries were sequenced on an Illumina HiSeq 2500 instrument (Illumina). Data were processed using RTA version 1.18.54 and 1.18.64, with default filter and quality settings. The reads were demultiplexed with CASAVA 1.8.4 and 2.17 (allowing 0 mismatches). Transcripts were mapped to the genome sequences (available from PomBase) using TopHat2 (Kim et al, 2013; Lock et al, 2019). HTSeq was used to count the number of reads per exon (Anders et al, 2015; Lock et al, 2019). The reads across exons were summed to obtain the total number of reads per gene. This procedure yielded raw counts $c_{ijk}$ for each gene $i$, growth medium $j$, and biological replicate $k$. Per sample normalisation was performed using the DESeq2 *estimateSizeFactors* function, yielding size factors $S_{jk}$ for each sample (Love et al, 2014). The normalised counts were calculated as follows:

$$n_{ijk} = \frac{c_{ijk}}{S_{jk}}. \tag{1}$$

Unless otherwise noted, RNA-Seq analyses were performed using these normalised counts, which enabled between-sample comparison of the expression of genes or sets of genes.

Transcripts were processed as mRNA if they were assigned protein-coding status by PomBase. Non-coding RNAs were defined as those with PomBase identifiers starting "SPNCRNA," which are antisense and lincRNAs. tRNA and rRNA were excluded because their repetitiveness and high abundance made them difficult to sequence with the traditional RNA-seq protocols used in this study.

## Proteomics

Cell pellets from 10 ml of each turbidostat culture was frozen in dry ice and stored at −80°C until sample preparation. Once thawed, cells were resuspended in lysis buffer (1% sodium deoxycholate and 1% ammonium bicarbonate). Lysis was performed in a FastPrep instrument (MP Biomedical) for five pulses at a speed of 6 m/s. Total cell extracts were treated with 5 mM tris(2-carboxyethyl) phosphine (TCEP) for 15 min at room temperature to reduce the disulphide bonds. An alkylation reaction was performed with the addition of 10 mM iodoacetamide for 30 min at 25°C in the dark. The reaction was quenched using 12 mM N-acetyl-cysteine for 10 min. The proteins were quantified using a BCA Protein Assay Reducing Agent Compatible kit (Thermo Fisher Scientific) and 100 µg of total protein was used for digestion. To improve the cleavage efficiency, protein extracts underwent a double digestion, first with Lys-C (Wako Chemicals) for 4 h at 37°C using a 1:200 (w/w) ratio, and then overnight with porcine trypsin at 37°C using a 1:100 (w/w) ratio. Digestion was stopped by lowering the pH with TFA at a final volume of 1%. The sodium deoxycholate precipitate formed because of the lowered pH was removed by centrifuging the samples at 4°C for 15 min at 14,000 rpm. The precipitated detergent was then discarded. The digested peptides were vacuum dried and stored at −80°C until required for analysis.

The protein digests were analysed by liquid chromatography-tandem mass spectrometry (LC-MS/MS) via an untargeted analysis approach using data-dependent acquisition (DDA) (Ducret et al, 1998). The raw MS data were analysed using MaxQuant (Cox & Mann, 2008) and applying the Label-free Quantification algorithm (Cox et al, 2014) for DDA data analysis.

Protein digests were reconstituted in 0.1% TFA and transferred to autosampler vials for LC-MS/MS analysis. The tryptic peptides were separated using an Ultimate 3000 RSLC nano liquid chromatography system (Thermo Fisher Scientific) coupled to a Q-Exactive tandem mass spectrometer (Thermo Fisher Scientific) via an EASY-Spray source. Sample volumes were loaded onto a trap column (Acclaim PepMap 100 C18, 100 µm × 2 cm) at 8 µl/min of 2% acetonitrile, 0.1% TFA. Peptides were eluted on-line to an analytical column (EASY-Spray PepMap C18, 75 µm × 75 cm). Peptides were separated at 200 nl/min with a ramped 180 min gradient using 4–30% buffer B (buffer A: 2% acetonitrile, 0.1% formic acid; buffer B: 80% acetonitrile, 0.1% formic acid) over 150 min, and 30–45% buffer B over 30 min. Eluted peptides were analysed by operating in positive polarity using a DDA mode. Ions for fragmentation were determined from an initial MS1 survey scan at 70,000 resolution (at m/z 200) in the Orbitrap followed by higher energy collisional dissociation (HCD) of the top 12 most abundant. MS1 and MS2 scan AGC targets set to $3 \times 10^6$ and $5 \times 10^4$ for maximum injection times of 50 and 110 ms, respectively. A survey scan covering the range of 400–1,800 m/z was used, with HCD parameters of isolation width 2.0 m/z and a normalised collision energy of 27%.

DDA data were processed using the MaxQuant software platform (v1.6.2.3) (Cox & Mann, 2008) with database searches performed by the in-built Andromeda search engine against the PomBase database (5,138 entries, v.20190507) (Lock et al, 2019). A reverse decoy database was created, and the results displayed at a 1% false discovery rate (fdr) for peptide spectrum matches and identified proteins. The search parameters included trypsin, two missed cleavages, fixed modification of cysteine carbamidomethylation, and variable modifications of methionine oxidation, asparagine deamidation, N-terminal glutamine to pyroglutamate modification, and protein N-terminal acetylation. Label-free quantification was enabled with an LFQ minimum ratio count of 2. The "match between runs" function was used with match and alignment time limits of 0.7 and 20 min, respectively.

Intensities were based on identified unique and razor peptides, and intensity-based absolute quantification (iBAQ) was calculated as the raw intensity/number of obtainable tryptic peptides. For the post-processing of the MaxQuant output, the data were filtered for detection in all three biological replicates. Subsequently, proteome mass fractions $\phi_{ij}$ were calculated for each protein group $i$, sample from growth medium $j$, replicate $k$ from the reported protein masses $m_i$, and the iBAQ quantities $B_{ijk}$ as follows:

$$\phi_{ijk} = \frac{m_i B_{ijk}}{\sum_l m_l B_{ljk}}. \tag{2}$$

## Repeated median linear models

As shown in the main text, several genes were enriched in one or more growth conditions in addition to growth rate correlations. The presence of such outliers affected the fit quality of the standard ordinary least squares linear model fits. To account for this, we used repeated median linear models (RMLM) for fitting regression lines (Siegel, 1982), as implemented in the R package "mblm." This method is robust when up to 50% of outliers are present in the data, and the working is described below.

In general, the data can be described as $N$ pairs of the growth rate $\mu$ and some expression value $y$ ($N = 24$ if expression was detected across all samples, or a smaller multiple of 3 when data were missing). From each observation $(\mu, y)_i$, a line is drawn to each of the other $N − 1$ points $(\mu, y)_j$, and the median slope and $y$-intercept of these $N − 1$ lines is associated with the data point $i$. The regression coefficients for the slope and $y$-intercept of the repeated median linear model are defined as the medians of all $N$ slopes and $y$-intercepts.

The regression slope and intercept are both proportional to the average expression level of a gene (protein). A fair comparison of the steepness of the growth rate dependencies between proteins or transcripts with different expression levels can therefore not use the regression parameters directly. To compare the growth law shape of protein groups with varying absolute abundances, the fold-change $FC$ was defined from the RMLM as the ratio

$$FC = \frac{y(\mu = \mu_{max}) - y(\mu = 0)}{y(\mu = 0.5\mu_{max})}, \tag{3}$$

with $\mu_{max} = 0.3$ h$^{-1}$. This can be expressed in terms of the fitted slope $a$ and the $y$-intercept $b$ as follows:

$$FC = \frac{\mu_{max}}{0.5\mu_{max} + b/a}. \tag{4}$$

## Hierarchical clustering

We used $z$-scores to normalise for variations in absolute expression levels. For each gene or protein group $i$ in the sample with medium $j$ and replicate $k$, the $z$-score was calculated as follows:

$$z_{ijk} = \frac{y_{ijk} - \mu_i}{\sigma_i}, \tag{5}$$

from the expression values $y_{ijk}$, where $\mu_i$ and $\sigma_i$ are the mean and standard deviations across all samples. The analysis was performed only on genes or protein groups that were detected across all 24 samples. The resulting matrices of the $z$-scores were analysed using hierarchical clustering and principal component analysis.

Hierarchical clustering on genes/protein groups was performed using the Euclidean distance and Ward linkage ("*ward.D2*"), using the "*hclust*" implementation of the R statistical language (v.3.5.3). In the transcriptome analysis, separate dendrograms were constructed for coding and non-coding RNAs, using the protein-coding list from PomBase and selecting ncRNAs from the presence of "NCRNA" in the systematic IDs.

### Sector assignment

For each gene or protein group $i$, we calculated R-squared ($R^2$), defined as follows:

$$R_i^2 = 1 - \frac{\sum_{j,k} r_{ijk}^2}{\sum_{j,k} (y_{ijk} - \mu_i)^2}, \tag{6}$$

and the associated P-values using the "summary.lm" method. Here, $r_{ijk}$ denotes the residuals from the RMLM fit, $y_{ijk}$ the expression (normalised counts or proteome fractions), $\mu_i$ the mean expression across samples, and the summation was performed across all $N$ samples where the gene was detected. We calculated the tail-based *fdr* (or *q*-values) and local *fdr* using the "fdrtool" R package and the false non-discovery rate cut-off method (Strimmer, 2008). Genes or protein groups were assigned to the P- or R-sector when their tail-based *fdr* < 0.1. R- and P-sector genes had positive and negative slopes, respectively, as determined by the fitted RMLM. In Table S6, hits with local *fdr* < 0.1 were flagged as confident.

To assess fit quality, in addition to $R^2$, we used a normalised sum of squared residuals, defined as follows:

$$SSR_{\text{norm},i} = \frac{1}{N-1} \frac{\sqrt{\sum_{j,k} r_{ijk}^2}}{\mu_i}, \tag{7}$$

with the notations as described in the previous paragraph.

### Protein-to-RNA ratios

For each protein group or gene $i$, and sample from growth medium $j$ and replicate $k$, the proteome number fraction was calculated from the iBAQ quantities $B_{ijk}$ as follows:

$$\psi_{P,ijk} = \frac{B_{ijk}}{\sum_l B_{ljk}}, \tag{8}$$

and the transcriptome number fraction was calculated from the normalised counts $n_{ijk}$ and the transcript lengths $l_i$ as follows:

$$\psi_{M,ijk} = \frac{\frac{n_{ijk}}{l_i}}{\sum_l \left(\frac{n_{ljk}}{l_i}\right)}. \tag{9}$$

These two number fractions are directly comparable. Note that the ratio $\psi_{P,ijk}/\psi_{M,ijk}$ is proportional to the ratio of absolute protein and mRNA numbers per cell (denoted $N_{P,ijk}$ and $N_{M,ijk}$, respectively) with a sample-specific normalisation:

$$\frac{\psi_{P,ijk}}{\psi_{M,ijk}} = \frac{N_{P,ijk}}{N_{M,ijk}} \frac{M_{jk}}{P_{jk}}, \tag{10}$$

where $M_{jk}$ and $P_{jk}$ denote the total number of transcripts and proteins per cell, respectively.

As seen in Fig S9, the protein-to-mRNA ratio is heavily dependent on the average expression level. To perform a meaningful analysis of between-sample protein-to-mRNA ratio differences, this effect was removed in the following way (Franks et al, 2017). The residual log-transformed protein-to-mRNA ratio was calculated as follows:

$$\text{residual } \log_2\left(\frac{\psi_{P,ijk}}{\psi_{M,ijk}}\right) = \log_2\left(\frac{\psi_{P,ijk}}{\psi_{M,ijk}}\right) - \underset{j,k}{\text{median}}\left[\log_2\left(\frac{\psi_{P,ijk}}{\psi_{M,ijk}}\right)\right]. \tag{11}$$

This way, the across-sample variation in protein-to-mRNA ratios could be compared between genes with wildly varying average expression levels, as in Fig 3B.

### Spearman correction

Noise in observations causes observed correlations between these observations to underrepresent true underlying correlations. However, a so-called Spearman correction can be used to mitigate this effect; it has previously been applied to omics data (Csárdi et al, 2015; Franks et al, 2017). Central to the method are the reliabilities $r_{P,j}$ and $r_{M,j}$ of the protein and mRNA data. For our data, these are defined for each condition $j$ as the geometric mean of observed pairwise Pearson correlations between the three biological replicates:

$$r_{P,j} = \sqrt[3]{\rho_{j,1:2}^P \rho_{j,1:3}^P \rho_{j,2:3}^P} \tag{12}$$

and

$$r_{M,j} = \sqrt[3]{\rho_{j,1:2}^M \rho_{j,1:3}^M \rho_{j,2:3}^M}. \tag{13}$$

Here, $\rho_{j,k1:k2}^P$ represents the Pearson correlation between the observed $\log_2$-transformed proteome number fractions in condition $j$ for replicates $k1$ and $k2$, and $\rho_{j,k1:k2}^M$ that for the correlations between $\log_2$-transformed transcriptome number fractions. Likewise, a first, uncorrected, estimate of the protein–mRNA correlation is calculated as the geometric mean of observed pairwise protein–mRNA correlations:

$$\hat{\rho}_j = \sqrt[6]{\rho_{j,1:1}\rho_{j,1:2}\rho_{j,1:3}\rho_{j,2:2}\rho_{j,2:3}\rho_{j,3:3}}. \tag{14}$$

The final, corrected, estimate of the protein–mRNA correlation is now given by the following equation:

$$R_j = \frac{\hat{\rho}_j}{\sqrt{r_{P,j}r_{M,j}}}. \tag{15}$$

## Gene set enrichment analysis

A multilevel gene set enrichment analysis was performed against the three *S. pombe* GO-slims (Gene Ontology Consortium, 2019; Lock et al, 2019) using the fgsea package (v1.16.0) (Korotkevich et al, 2021 *Preprint*), with boundary parameter $\varepsilon = 0$. Genes (protein groups in the proteomics analysis) were ranked based on the following signed measure of significance:

$$-\text{sign}(a_i)\log_{10}(p_i), \tag{16}$$

Here, $a_i$ and $p_i$ are the slope and *P*-value associated with the RMLM growth rate fit for gene (protein group) $i$, and sign $(a_i)$ equals 1, 0, or −1 when $a_i$ is positive, zero, or negative, respectively. This resulted in a list where the R-sector was at the top of the list, and the P-sector at the bottom. Unlike the traditional functional enrichment, however, this analysis does not require an arbitrary cut-off point.

## Bootstrapping

For the analysis illustrated in Figs 5A and B, S14, and S17, 1,000 bootstrap samples were generated using the *bootstraps* function from the "rsample" package (v0.0.8). The RMLM analysis was repeated on the bootstrapped samples, resulting in sample distributions for the RMLM slopes, intercepts, and *FCs*. Plots of the 2.5–97.5% confidence interval were drawn using the RMLM predictions on a 101-point grid spanning 0–0.3 h$^{-1}$.

Other confidence intervals were drawn using the *geom_smooth* function in ggplot2 (v3.3.2) (Wickham, 2016) with the default 95% confidence interval and the RMLM method, unless otherwise noted.

## Barcode plots

For the barcode plots in Figs S23 and 6G and H, the directed length $l_{ij}$ of the bar for protein $i$ and medium $j$ was calculated from the median proteome mass fractions across the three biological replicates,

$$x_{ij} = \underset{k=1,2,3}{\text{median}} \phi_{ijk}, \tag{17}$$

and the median across all samples,

$$M_i = \underset{j,k}{\text{median}} \phi_{ijk}, \tag{18}$$

in the following way:

$$l_{ij} = \frac{x_{ij} - M_i}{M_i}, \tag{19}$$

with missing data imputed to zero. The scale was capped at $-1 < l_{ij} < 2$.

## Differential expression analysis

To identify differential expression in the transcriptome on top of growth-rate-mediated effects, we performed an analysis using "DESeq2" (v1.22.2) from the Bioconductor suite (v3.8) (Love et al, 2014; Huber et al, 2015), comparing the residual expression in each condition to a synthetic reference condition. The fold change obtained by this procedure can be interpreted as the ratio of observed normalised counts and the counts predicted by the RMLM, and the associated *P*-value provides an interpretable estimate of significance.

The DESeq2 analysis pipeline enables the introduction of per-gene, per-sample normalisation factors that are commonly used to correct for batch-dependent GC-content or length biases. We adapted this functionality to normalise the growth rate bias of each gene, by introducing factors $N_{ijk}$ that converted between the measured raw counts $c_{ijk}$ and RMLM-predicted raw counts $q_{ijk}$:

$$q_{ijk} = \frac{c_{ijk}}{N_{ijk}}, \tag{20}$$

in analogy to the definition of size factors in Equation (1). However, the fitting of RMLMs yielded per-gene, per-sample predictions $p_{ijk}$ of the normalised counts. Using the sample-dependent size factors, we converted these to predictions of raw counts as follows:

$$q_{ijk} = p_{ijk}S_{jk}. \tag{21}$$

Therefore, the normalisation factors were calculated as follows:

$$N_{ijk} = \frac{c_{ijk}}{S_{jk}p_{ijk}} = \frac{n_{ijk}}{p_{ijk}}. \tag{22}$$

We excluded genes with negative predicted raw counts and rescaled the normalisation factors across samples for each gene to have a geometric mean of 1 for numerical accuracy.

Using the RMLM-predicted raw counts, we further defined a synthetic reference condition with three biological replicates by using the median predicted count across all growth media for each replicate as follows:

$$s_{ik} = \text{int}\left(\underset{j}{\text{median}} \, q_{ijk}\right). \tag{23}$$

These reference counts were rounded to the nearest integer, as they represent raw counts in the DESeq2 pipeline. By design, the $q_{ijk}$ have no residual growth rate trend.

Subsequently, the analysis proceeded on the constructed data set with nine conditions: the original eight and the synthetic one, with each set having three biological replicates. Pairwise fold-changes $F$ and the associated *P*-values (both uncorrected and adjusted $p_{\text{adj}}$) are reported between the eight growth media and the synthetic reference. Fold-changes were shrunk using the *lfcShrink* function of DESeq2, using the "apeglm" method (Love et al, 2014; Zhu et al, 2019). Genes were reported as differentially expressed (DE) if $p_{\text{adj}} < 0.01$ and $|\log_2 F| > 0.5$ for at least one condition.

## Functional enrichment

We performed one-sided Fisher's exact tests to assess the enrichment of DE genes across the *S. pombe* GO-slims and terms from

the biological_process GO with at most 50 annotations in *S. pombe* (Gene Ontology Consortium, 2019; Lock et al, 2019). From the resulting *P*-values, local false discovery rates (*lfdr*) were calculated using the "fdrtool"s false non-discovery rate method (Strimmer, 2008).

In the enrichment plots for the GO-slim terms (Figs 7B and S1), terms with *lfdr* < 0.05 were deemed significant, and the terms were ordered from top to bottom by increasing the smallest *lfdr* to aid interpretation. For the biological_process enrichment plot (Fig S28), the significance threshold was *local fdr* < 0.001. The significant terms were clustered hierarchically using the Euclidean distance and Ward linkage ("*ward.D2*"), using the "*hclust*" implementation of the R statistical language (v.3.5.3).The terms were ordered by the smallest *lfdr* as much as possible, while remaining consistent with the clustering constraint.

## Data Availability

Proteomics data are deposited in PRIDE (PXD027835) and RNA-Seq data in ArrayExpress (E-MTAB-10778). For the purpose of open access, the authors have applied a CC BY public copyright licence to any Author Accepted Manuscript version arising from this submission.

## Supplementary Information

## Acknowledgements

We would like to thank Xi-Ming Sun, Lucie Martin, Wenhao Tang, Benjamin Heineike, Jürg Bähler, Juan Mata, and Stephan Kamrad for critical reading of the manuscript, as well as Pranas Grigaitis, Eunice van Pelt-Kleinjan, and the Shahrezaei, Marguerat and Ralser labs for discussion and feedback. We would like to thank the MRC London Institute of Medical Sciences Genomics facility for help with sequencing. IT Kleijn was supported by the Wellcome Trust (203968/Z/16/Z). F Bertaux received financial support from Leverhulme Research Project Grant (RPG-2014-408) awarded to S Marguerat and V Shahrezaei. V Shahrezaei is supported by the EPSRC Centre for Mathematics of Precision Healthcare (EP/N014529/1). A Martínez-Segura, M Saint, H Kramer, and S Marguerat are supported by the UK Medical Research Council.

### Author Contributions

IT Kleijn: conceptualization, data curation, formal analysis, investigation, visualization, and writing—original draft, review, and editing.
A Martinez-Segura: conceptualization, data curation, formal analysis, investigation, and writing—review and editing.
F Bertaux: conceptualization, investigation, and writing—review and editing.
M Saint: conceptualization, investigation, and writing—review and editing.
H Kramer: investigation.
V Shahrezaei: conceptualization, supervision, funding acquisition, project administration, and writing—original draft, review, and editing.
S Marguerat: conceptualization, formal analysis, supervision, funding acquisition, project administration, and writing—original draft, review, and editing.

### Conflict of Interest Statement

The authors declare that they have no conflict of interest.

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
