## [Reviewer comments · Life Science Alliance]

Life Science Alliance

GROWTH-RATE DEPENDENT AND NUTRIENT-SPECIFIC GENE EXPRESSION RESOURCE ALLOCATION IN FISSION YEAST

Istvan Kleijn, Amalia Martínez-Segura, François Bertaux, Malika Saint, Holger Kramer, Vahid Shahrezaei, and Samuel Marguerat

DOI: <https://doi.org/10.26508/lsa.202101223>

Corresponding author(s): Samuel Marguerat, Imperial College London and Vahid Shahrezaei, Imperial College London

Review Timeline:

Submission Date:	2021-09-01
Editorial Decision:	2021-09-02
Revision Received:	2021-12-06
Editorial Decision:	2022-01-10
Revision Received:	2022-01-25
Accepted:	2022-01-25

Transaction Report:

Please note that the manuscript was reviewed at Review Commons and these reports were taken into account in the decision-making process at Life Science Alliance.

Review
COMMONS

Review Reports from Review Commons

Review #1

In this paper Kleijn et al study global gene expression profiles in *S. pombe* grown in different nitrogen sources using a turbidostat that result in variation in growth rate.

The authors use both RNAseq to quantify RNA expression and mass spectrometry to quantify protein expression. They find that the expression of many genes is correlated with growth rate. This finding builds on prior work performed by other groups in *S. cerevisiae* and bacteria that show organismal growth rate is a primary determinant of gene expression state for a large fraction of genes. The findings in this paper confirm and extend those results.

One surprising aspect of this manuscript is that the authors do not seem to have made the most of their experimental design. The acquisition of both protein and mRNA expression across these conditions provides a unique dataset for looking at how these two levels of expression agree with respect to each other. A simple plot showing the strength of the growth rate response for a gene at the level of mRNA and protein would already be interesting, but I would think that there is the opportunity to look more quantitatively at whether the ratio of mRNA to protein remains constant across growth rates or whether there systematic deviations that are biologically interesting. I would encourage the authors to address this question with their unique dataset.

Prior to publication the authors should address the following points.

At what point in the turbidostat cycle was the sampling performed? At steady state or during the dilution phase?

It is unclear in the text what transcripts are included in the category ncRNA. Does this include tRNA and rRNA?

The basis for the abbreviations for positive (R) negative (P) and not significant (Q) are obscure. Why not P, N, NS?

In Brauer et al., the fraction of cells in G1 is correlated with growth rate. Is that the case in *S pombe*? Is there any relationship between cell cycle gene expression and growth rate related gene expression?

Is there anything unique to the set of ~100 genes that are anticorrelated between mRNA and protein in response to growth rate variation?

A clearer explanation of the FC metric and the rationale for its use should be made in the results. What is FC an abbreviation for? It is unclear why this metric is needed, when the strength of the response to growth rate is captured by the slope.

Airoldi et al., 2016 and Airoldi et al., 2009 looked at methods for normalizing gene expression to growth rate and may be relevant sources.

The contrast in the experimental rationale between using chemostats and turbidostats is interesting, but I am left unclear about whether the result is really that different. What is the key distinction in the observed data in comparing gene expression response to growth rate in the chemostat and turidostat?

Review #2

Kleijn et al. measured transcript and protein abundance in fission yeast cultures growing on different nutrient sources (and thus at different growth rates) in turbidostats. Their experimental design is sound and the data quality appears good. The authors focus on analyzing their data from the vantage point of previously reported ideas on principles of proteome allocation, and expand beyond this framework with interesting analyses, e.g., on the stoichiometry of translation complexes changes with the growth rate.

Generally I find the paper well written and the conclusions well substantiated. Below are specific recommendations that may help the authors improve their study:

- Your data allow investigating the extend of transcriptional and post-transcriptional regulation in fission yeast, and I think this analysis will be very interesting. PMID: 28481885 provides one simple approach to such analysis, and the

authors may use another. Importantly, they authors must account for measurement noise.

- Your analysis of the ribosomal proteins (RP), the ribosome biogenesis regulon (RiBi), and the translation initiation, elongation and termination factors (IET) is interesting. I would love to know whether there changes within these groups of proteins, e.g., different RP in budding yeast change differently with growth rate (PMID: 24767987, PMID: 26565899) and I would love to know if this is the case with fission yeast.

- The Z score ranges on some of the heatmaps (e.g. fig 2A) are so wide that the changes in protein / RNA abundance are difficult to see.

- It will be very useful to perform unbiased gene set enrichment analysis of the functions that show significant growth rate dependent and nutrient dependent effects, e.g., as in Fig 11 of PMID: 21525243

I am an expert in this field, and I think that this study represents a significant advance.

Manuscript number: RC-2021-00724R

Corresponding author(s): Samuel, Marguerat and Vahid Shahrezaei

[The “revision plan” should delineate the revisions that authors intend to carry out in response to the points raised by the referees. It also provides the authors with the opportunity to explain their view of the paper and of the referee reports.]

The document is important for the editors of affiliate journals when they make a first decision on the transferred manuscript. It will also be useful to readers of the reprint and help them to obtain a balanced view of the paper.

*If you wish to submit a full revision, please use our "Full Revision" template. **It is important to use the appropriate template to clearly inform the editors of your intentions.**]*

1. General Statements [optional]

This section is optional. Insert here any general statements you wish to make about the goal of the study or about the reviews.

We acknowledge the constructive reviews of our paper obtained through Review Commons. We are confident that we can address all the reviewer comments and we believe this additional analysis will enhance significantly the impact of our paper. We provide below a point-to-point plan for our revisions. It also includes a draft of a new figure 3 and accompanying analysis we propose to add to the manuscript. In addition, we are including a note that explains a minor change in our proteomic data due to a pre-processing error, which does not affect our results and conclusions.

2. Description of the planned revisions

Insert here a point-by-point reply that explains what revisions, additional experimentations and analyses are planned to address the points raised by the referees.

Reanalysis of the proteomics dataset:

While preparing the raw proteomics data for deposition in PRIDE, we have discovered an error in the original MaxQuant DDA processing of the dataset (original submission: p. 40, lines 25–34). The dataset used in the manuscript submitted to Review Commons contained two additional replicates of cells grown in ammonium chloride (Amm). The samples were not included in the final processed dataset as they lacked corresponding growth curves and RNA-seq data. Because the mass spectrometry analysis matches detected peaks across all samples, the presence of additional replicates during the early analysis steps affected peptide detection. For consistency, we have

September 2, 2021

Re: Life Science Alliance manuscript #LSA-2021-01223

Samuel Marguerat
Imperial College London

Dear Dr. Marguerat,

Thank you for submitting your manuscript entitled "GROWTH-RATE DEPENDENT AND NUTRIENT-SPECIFIC GENE EXPRESSION RESOURCE ALLOCATION IN FISSION YEAST" to Life Science Alliance. We invite you to re-submit the manuscript, revised according to your proposed Revision Plan.

Thank you for this interesting contribution to Life Science Alliance. We are looking forward to receiving your revised manuscript.

Sincerely,

B. MANUSCRIPT ORGANIZATION AND FORMATTING:

Dear Eric,

Please find attached our revised manuscript that we would like to submit for publication in Life Science Alliance following your invitation.

We thank the reviewers of the Review Commons Submission. In this revised submission we have followed the plans we presented to you based on reviewer's comments and have fully revised our paper with some new analysis that has greatly improved our paper. Also, as mentioned in our revised plan, we have corrected some minor inconsistencies in the processing of the data that we noticed after the original submission to Review Commons.

In our rebuttal, the changes proposed in our revision plan are highlighted in blue. In black you can find what has actually been changed in line with the proposals. Reviewers' comments are in bold red. A summary of all the changes is also presented in a table at the end of this rebuttal letter.

While preparing the raw proteomics data for deposition in PRIDE, we have discovered an error in the original MaxQuant DDA processing of the dataset (original submission: p. 40, lines 25–34). The dataset used in the manuscript submitted to Review Commons contained two additional replicates of cells grown in ammonium chloride (Amm). The samples were not included in the final processed dataset as they lacked corresponding growth curves and RNA-seq data. Because the mass spectrometry analysis matches detected peaks across all samples, the presence of additional replicates during the early analysis steps affected peptide detection. For consistency, we have decided to reanalyse the data including only the 24 Proteomics and RNA-seq matched samples from this manuscript.

Specifically, iBAQ values were only marginally affected by the reanalysis and only a small number of proteins were not found in both original and reanalysed datasets. This affected mainly the quantification of a few lowly expressed proteins and not the expression of abundant proteins, or of groups of proteins such as the functional classes analysed in this manuscript. Only one minor conclusion from the original submission had to be modified: the fold change of the ribosome biogenesis growth rate is not significantly different from that of the ribosomal proteins (p. 10, l. 10-12). We apologise for this mistake and will include new figures in the revised manuscript. Again, this will not change the figures significantly and the conclusion of our work remain unaffected. We include an updated figure 2, which is the overall description of the dataset as an example. While the numbering of the clusters has changed, the conclusions drawn were not affected.

This has been addressed in the text and all figures and tables in the revised submission have been regenerated to reflect the changes in the underlying data. In total, 58 fewer protein groups were detected in the re-analysed data set. Proportions of proteins assigned to lists such as P/Q/R sector and the WFSP proteins were affected by at most two percentage points either way and the text has been adjusted.

Reviewer #1 (Evidence, reproducibility and clarity (Required)):

In this paper Kleijn et al study global gene expression profiles in *S. pombe* grown in different nitrogen sources using a turbidostat that result in variation in growth rate. The authors use both RNAseq to quantify RNA expression and mass spectrometry to quantify protein expression. They find that the expression of many genes is correlated with growth rate. This finding builds on prior work performed by other groups in *S. cerevisiae* and bacteria that show organismal growth rate is a primary determinant of gene expression state for a large fraction

of genes. The findings in this paper confirm and extend those results.

Reviewer #1 (Significance (Required)):

One surprising aspect of this manuscript is that the authors do not seem to have made the most of their experimental design. The acquisition of both protein and mRNA expression across these conditions provides a unique dataset for looking at how these two levels of expression agree with respect to each other. A simple plot showing the strength of the growth rate response for a gene at the level of mRNA and protein would already be interesting, but I would think that there is the opportunity to look more quantitatively at whether the ratio of mRNA to protein remains constant across growth rates or whether there systematic deviations that are biologically interesting. I would encourage the authors to address this question with their unique dataset.

This point has been raised by both reviewers and we acknowledge that this is an important aspect of our dataset. We have taken the reviewer suggestions on board and will add a new main figure 3 as well as several supplementary figures to our manuscript (draft main figure and caption below). Panel C shows the plot suggested by the reviewer in this specific section of the review (“showing the strength of the growth rate response for a gene at the level of mRNA and protein”). It reveals that, while there is overall a significant correlation between mRNA and protein, this is not perfect and there are evidence of post-transcriptional regulation of gene expression across growth conditions. This will be discussed in details in the revised manuscript.

This has now been addressed in detail in the text (pp. 9-10) and Methods (pp. 49-51) as well as by the addition of 1 new main text figure (Fig 3), 7 Supp Figs (S6-S12) and 6 Supp Tables (S7-S12).

Prior to publication the authors should address the following points.

At what point in the turbidostat cycle was the sampling performed? At steady state or during the dilution phase?

Cultures were harvested at the very end of the turbidostat growth curves as presented in Figure 1B. This is outside of the dilution phase. We will indicate this on Figure 1B with an arrow.

Figure 1B and the text (p. 7) have been adjusted to clarify this.

It is unclear in the text what transcripts are included in the category ncRNA. Does this include tRNA and rRNA?

Only transcripts annotated as SPNCRNA in the fission yeast genome were included. These are antisense and lincRNAs. tRNA and rRNA were not included as they are repetitive and difficult to sequence with traditional RNA-seq protocols. We will clarify this in the revised text.

A clarifying sentence has been added to the text (p. 7).

The basis for the abbreviations for positive (R) negative (P) and not significant (Q) are obscure. Why not P, N, NS?

This follows the convention notation from Scott et al (Science 2010). This terminology is well accepted in bacteria and yeast and we decided to use it for consistency. We will make this clear and explain the origin of abbreviations in the revised text.

This has been rephrased in the introduction and the origin of R sector in ribosomal proteins has been added (p. 3).

In Brauer et al., the fraction of cells in G1 is correlated with growth rate. Is that the case in S pombe? Is there any relationship between cell cycle gene expression and growth rate related gene expression?

Fission yeast cell cycle is a bit peculiar in that the main growth phase occurs in G2 which represent ~70% of the total cell cycle. When we looked at cell cycle marker we did not observe a simple relationship between cell cycle markers at the protein and mRNA level and the growth rate (see figure S4.5C-E). This is in line with flowcytometry and microscopy data published in Carlson et al (JCS 1999, 112:939) which also failed to detect a straightforward relationship between length of cell cycle phases and the growth rate upon growth on different amino acid sources. We will discuss this point in the revised manuscript.

This point has been added, see p. 14.

Is there anything unique to the set of ~100 genes that are anticorrelated between mRNA and protein in response to growth rate variation?

We have added a panel in the new Figure 3 to address this point. In summary, those genes are not significantly enriched for specific functional categories. We will discuss single gene in the revised manuscript.

This analysis is reported in Figure 3C and Supp Fig S11. A minor enrichment for proteasomal genes was found in the class with positive growth correlations in the transcriptome and negative ones in the proteome.

A clearer explanation of the FC metric and the rationale for its use should be made in the results. What is FC an abbreviation for? It is unclear why this metric is needed, when the strength of the response to growth rate is captured by the slope.

The regression slope between protein or RNA amounts and the growth rate is proportional to their expression levels. To fairly compare regression slopes between proteins or RNA with different expression levels, this has to be normalised out. The FC value (Fold Change across the range of growth rates) has been designed to achieve this.

$$FC = (\text{spread of expression level between zero growth and maximal growth}) / (\text{median expression}).$$

We will extend the methods section and we will add a description of this measure in the main text for clarification.

The text has been clarified on p. 8 and Methods, p. 47.

Airoldi et al., 2016 and Airoldi et al., 2009 looked at methods for normalizing gene expression to growth rate and may be relevant sources.

This is an interesting suggestion. We will explore the possibility of using analysis of covariance (ANCOVA) of a joint growth-rate/nutrient-specific model to strengthen our claims about the respective contribution of growth-related and condition-specific regulation. We suspect that the required assumption of homogeneity of regression slopes may be not be met in our dataset but will formally look into it.

See next point.

The contrast in the experimental rationale between using chemostats and turbidostats is interesting, but I am left unclear about whether the result is really that different. What is the key distinction in the observed data in comparing gene expression response to growth rate in the chemostat and turidostat?

The key distinction between chemostat and turbidostat is the metabolic limitation, which is externally imposed by limited nutrient *quantity* in the chemostat, and internally by limited nutrient *quality* in the turbidostat. In other words, the turbidostat functions as a continuous flask culture where it is possible to determine the effect of medium composition, rather than varying the concentration of a single limiting nutrient. The key difference in our data resides in the expression of the metabolic enzymes which is not as clearly anticorrelated with the growth rate as in the case of nutrient limitation in turbidostat. As pointed out by the reviewer, this is an important aspect of our study and we will make it clearer in the revised manuscript.

We realised that this point and the previous point are related. Because there is almost no variance in growth rates between replicates in the same growth medium, the suggested Airoidi et al papers do not apply to our turbidostat experiments. Rather, these methods are applicable to chemostat cultures, where each limiting nutrient is perturbed to yield a range of different growth rates. The text has been expanded to point this out (p. 19).

Reviewer #2 (Evidence, reproducibility and clarity (Required)):

Kleijn et al. measured transcript and protein abundance in fission yeast cultures growing on different nutrient sources (and thus at different growth rates) in turbidostats. Their experimental design is sound and the data quality appears good. The authors focus on analyzing their data from the vantage point of previously reported ideas on principles of proteome allocation, and expand beyond this framework with interesting analyses, e.g., on the stoichiometry of translation complexes changes with the growth rate.

Generally I find the paper well written and the conclusions well substantiated. Below are specific recommendations that may help the authors improve their study:

- Your data allow investigating the extend of transcriptional and post-transcriptional regulation in fission yeast, and I think this analysis will be very interesting. PMID: 28481885 provides one simple approach to such analysis, and the authors may use another. Importantly, they authors must account for measurement noise.

As mentioned in response to reviewer one, we have made a new figure in address this issue (see the draft of the new figure 3 above). In particular, we have used the method suggested in the reference mentioned in this comment to quantify significance of the correlations observed between mRNA and protein levels (panel A in the new figure 3).

This is now discussed in the text (pp. 9-10 and Methods, pp. 49-51). The Spearman correction has been applied to account for measurement noise, and gene-wise protein-to-mRNA ratios have been calculated using the normalisation method described in the paper the reviewer helpfully suggested.

- Your analysis of the ribosomal proteins (RP), the ribosome biogenesis regulon (RiBi), and the translation initiation, elongation and termination factors (IET) is interesting. I would love to know whether there changes within these groups of proteins, e.g., different RP in budding yeast change

differently with growth rate (PMID: 24767987, PMID: 26565899) and I would love to know if this is the case with fission yeast.

This is a very intriguing and timely question. We had superficially touched upon this on Figure S4.2. On panel S4.2A for instance we report ribosomal proteins (RP) and translation factors that are not part of the R sector (no expression correlation with growth). We agree with the reviewer that extending this analysis would add to the paper. We will, therefore, expand this figure and include profiles of specific RPs and translation factors asking whether their expression is condition specific. We will also compare and contrast these findings with RPs orthologues in *S. cerevisiae* and if possible *E. coli*.

An additional supplementary figure (S16) has been added with the expression profiles of the RPs that were not significantly positively correlated with the growth rate, and particular examples and their orthologues are discussed in the text (p. 12).

- The Z score ranges on some of the heatmaps (e.g. fig 2A) are so wide that the changes in protein / RNA abundance are difficult to see.

The plotted z-scores in Fig 2A have been truncated to range from -3 to +3, showing differences in expression more clearly (see updated figure 2 below). We will do this to the other figures too.

The truncation has been applied to the heatmaps in Fig 2 and Supp Figs S2 and S3, indeed showing the changes in gene expression more clearly.

- It will be very useful to perform unbiased gene set enrichment analysis of the functions that show significant growth rate dependent and nutrient dependent effects, e.g., as in Fig 11 of PMID: 21525243

We thank the reviewer for this suggestion and we will perform gene set enrichment analyses (GSEA). A preliminary analysis was performed using the `fgsea` R package on the growth rate correlations in the transcriptome, ranking the genes by their RMLM p-value. We will extend this to the growth-rate correlated proteins. Furthermore, we will analyse the main nutrient-dependent effect by utilising the correlations with the WFSP principal component.

We applied unbiased gene set enrichment analysis (GSEA) to the growth rate correlations of both the proteome and the transcriptome. The results of this are summarised in Fig 3D and the accompanying text on p. 10. The WFSP principal component was somewhat correlated with the growth rate (Fig 6E, Supp Fig S24) and a GSEA analysis of the WFSP signature was therefore not informative. However, we were able to apply the unbiased GSEAs to the medium-specific expression, ranking genes based on their fold-change difference between expression in each growth medium and the synthetic medium representing the growth-normalised expression. The results, shown in Supp Fig S29, did not yield much biological insight over the overlap analysis reported in Fig 7C and Supp Fig 28. More details on these analyses are provided in the Methods section, pp. 50-51.

In the case of the medium-specific expression after removal of the growth rate effect, the ranking-based GSEA was performed using corrected fold-changes after shrinkage. The reason for this was that rankings based on raw fold-change measures are biased towards lowly-expressed genes at their top and bottom ends. For consistency, we redefined the medium-specific signatures R1-R10 to account for this as well. The composition and numbering of the signatures changed slightly, and the text, Figure 7, and Supp Figs S27 and S28 have been adjusted to use the shrinkage-corrected fold-changes. The conclusions drawn from the analysis were not affected.

Reviewer #2 (Significance (Required)):

I am an expert in this field, and I think that this study represents a significant advance.

One further edit was made that was not requested by the reviewers. A minor error in Supp Figs S25 and S26 has been corrected so that the confidence intervals for the growth rate fits are shown for the entire range of growth rates plotted.

Lastly, the abstract has been shortened to fit within the word limit. Supplementary Tables, Figures, and Supplementary Figures have now all been numbered sequentially, and the figure font has been changed.

We provide here a table of the main changes made to the manuscript, with their places in the text, and the new and/or edited figures and tables:

	Text	Figures	Tables
I. Reanalysis of the proteomics dataset		Throughout	Throughout
Number of protein groups with significant variance across conditions	p. 7		
Proteome cluster numbering and (lack of) enrichment	pp. 7-8		
Numbers of protein groups in P/Q/R sector	pp. 8-9		
RiBi not significantly different from RP and IET	p. 11		
Numbers of protein groups in WFSP lists	p. 16		
II. Joint analysis of mRNA and protein expression	pp. 9-10, Methods pp. 49-51		
Spearman-corrected estimate of per-condition overall RNA-Protein correlation		3A, S6-S9	S7
Gene-specific RNA-Protein ratios after median subtraction		3B, S10	S8
Growth-rate correlations agreement (in FC)		3C	
GSEA of growth correlations		3D, S11-S12	S9-S12
III. RPs not in R sector	p. 12	S16	
IV. GSEA of medium-specific expression after growth-rate removal		S29	
New analysis	p. 18	S29	S19
Using LFC shrinkage redefined signatures, small changes in composition and numbering	p. 18	7, S27-S29	
V. Minor edits			
R-sector linked to ribosomes in introduction	p. 3		
Culture harvesting clarification	p. 7	1B	
Non-coding RNA clarification	p. 7		
Fold-change definition justified	p. 8, Methods p. 30		
Cell cycle markers	p. 14		
Chemostats vs turbidostats	p. 19		
Truncated heatmap colour scales		2A, S2-S3	
Repaired confidence intervals		S25-S26	

January 10, 2022

RE: Life Science Alliance Manuscript #LSA-2021-01223R

Dr. Samuel Marguerat
Imperial College London
London W12 0NN
United Kingdom

Dear Dr. Marguerat,

Thank you for submitting your revised manuscript entitled "GROWTH-RATE DEPENDENT AND NUTRIENT-SPECIFIC GENE EXPRESSION RESOURCE ALLOCATION IN FISSION YEAST". We would be happy to publish your paper in Life Science Alliance pending final revisions necessary to meet our formatting guidelines. Please address Reviewer 1's remaining comments.

- please upload your main manuscript text as an editable doc file
- please upload your Tables in editable excel format as separate files
- please upload your main and supplementary figures as single files
- please add the Twitter handle of your host institute/organization as well as your own or/and one of the authors in our system
- please add an Author Contributions section to your main manuscript text
- we encourage you to revise the figure legends for figures S2, S3 such that the figure panels are introduced in alphabetical order
- please consult our manuscript preparation guidelines <https://www.life-science-alliance.org/manuscript-prep> and make sure your manuscript sections are in the correct order
- please add callouts for Figures 1A; S2A, B; S3C, D; S4A-D; S8; S16C; S19E; S22 and S24A-D to your main manuscript text

A. FINAL FILES:

B. MANUSCRIPT ORGANIZATION AND FORMATTING:

Sincerely,

Reviewer #1 (Comments to the Authors (Required)):

In this resubmission the authors have addressed all of my comments. The additional analyses comparing mRNA and protein expression strengthens the paper considerably. This study is an important contribution to our understanding of cell growth regulation and gene expression.

In a final version the authors may wish to consider:

1. The authors define FC as the "spread of expression level...". Isn't the "difference" more precise than "spread".
2. All of the data are relative data. Is there any evidence that total mRNA or total proteome sizes are increasing with increased growth rate? If there are no data to address this, then it may simply be worth pointing out in the discussion.
3. The slope for the RP mass fraction in Figure 5D looks identical for cerevisiae and pombe. Therefore, it is difficult to understand the statement that "fission yeast used its RPs significantly more efficiently than the budding yeast". To me, the fact that the slopes are identical is a striking observation about the conservation between the two species. It is confusing about why there are data points for pombe and E coli, but not S cerevisiae.

January 25, 2022

RE: Life Science Alliance Manuscript #LSA-2021-01223RR

Dr. Samuel Marguerat
Imperial College London
du Cane Rd
London W12 0NN
United Kingdom

Dear Dr. Marguerat,

Thank you for submitting your Research Article entitled "GROWTH-RATE DEPENDENT AND NUTRIENT-SPECIFIC GENE EXPRESSION RESOURCE ALLOCATION IN FISSION YEAST". It is a pleasure to let you know that your manuscript is now accepted for publication in Life Science Alliance. Congratulations on this interesting work.

DISTRIBUTION OF MATERIALS:

Again, congratulations on a very nice paper. I hope you found the review process to be constructive and are pleased with how the manuscript was handled editorially. We look forward to future exciting submissions from your lab.

Sincerely,
